# Tumor-infiltrating immature innate lymphoid cells in colorectal cancer are biased toward ILC1/tissue-resident NK cell differentiation

Anne Marchalot [1], Malin Ljunggren[2], Christopher Stamper[1], Whitney Weigel[1], Christopher Andrew Tibbitt[1], Isabel Meininger[1], Ram Vinay Pandey[1], Miriam Franklin [1], John Washington Bassett[1], Lorenz Wirth[1], Colorectal Study Group*, Ulrik Lindforss[2], Gabriella Jansson-Palmer[2], Caroline Nordenvall[2,4] & Jenny Mjösberg [1,3,4] ✉

Peritoneal metastases (PM) occur in 10% of patients with colorectal cancer (CRC) and are linked to poor outcomes. Although dysregulated innate lymphoid cells (ILC) have been described in CRC, their function in CRC-PM remains unclear. Here, we analyze tumor samples from CRC and CRC-PM patients using single-cell RNA sequencing (11 patients), flow cytometry (8 patients) and differentiation assays (24 patients). Healthy colon, primary CRC and CRC-PM tumors are infiltrated by heterogeneous populations of ILC3, ILC2, ILC1, tissue resident (tr)NK cells and conventional (c)NK cells. Compared to healthy colons, primary CRC and CRC-PM tumors are depleted of ILC3 but enriched for ILC1, trNK cells and cNK cells. CRC and CRC-PM tumors harbor two immature ILC populations, early NK and naïve (n)ILC, with nILCs being transcriptionally skewed toward ILC1 and trNK cells. Indeed, co-culture of isolated nILCs with OP9-DL1 cells induces intratumoral nILC differentiation into ILC1/trNK-like cells. These findings help understand the immune pathogenesis of CRC and CRC-PM and provide insights for future ILC1 and NK cell-based therapies.

Colorectal cancer (CRC) is the second leading cause of cancer-related death globally[1]. Around 5% of CRC patients present with peritoneal metastasis (PM) at diagnosis[2,3], and around 5% of CRC patients develop PM later[3,4]. Patients with right-sided colon cancer, locally advanced CRC, especially those with obstructive or perforated tumors, or those undergoing emergency or non-radical resections, are at an even greater risk[3]. Historically, CRC-PM patients were treated palliatively, but now hyperthermic intraperitoneal chemotherapy (HIPEC)

combined with cytoreductive surgery (CRS) offers a potential cure for a subgroup of patients[5]. However, HIPEC efficacy is debated[6] and up to 25% of patients experience serious postoperative complications, and the risk of recurrence is high[7]. Immune checkpoint blockade has shown encouraging results for patients with microsatellite instability high (MSI-high) metastatic CRC[8], but only around 8% of patients with metastatic CRC have MSI-high tumors[9]. Hence, more research is needed on immunotherapies for patients with microsatellite stable (MSS)

[1]Department of Medicine Huddinge, Center for Infectious Medicine, Karolinska Institutet, Karolinska University Hospital Huddinge, Stockholm, Sweden. [2]Department of Molecular Medicine and Surgery, Karolinska Institutet, Stockholm, Sweden. [3] Department of Medicine Huddinge, Clinical Lung- and Allergy Research Unit, Karolinska Institutet, Stockholm, Sweden. [4]These authors contributed equally: Caroline Nordenvall, Jenny Mjösberg. *A list of authors and their affiliations appears at the end of the paper. ✉e-mail: jenny.mjosberg@ki.se

## Table 1 | Patient characteristics of samples used for single-cell RNA-sequencing

| Characteristic | Subgroup | Primary colon tumors N = 6 (%) | Peritoneal metastasis N = 7 (%) |
|---|---|---|---|
| Biological sex | Female (%) | 2 (33) | 2 (29) |
| | Male (%) | 4 (6%) | 5 (71) |
| Age at surgery | Mean years (range) | 66 (34–90) | 59 (34–72) |
| Tissues donated[a] | Normal colon (%) | 6 (100) | 2 (29) |
| | Colorectal primary tumor (%) | 6 (100) | 2 (29) |
| | Peritoneal metastasis (%) | 2 (33) | 7 (100) |
| American society of anesthesiologists status score | 1 | 3 (50) | 1 (14) |
| | 2 | 2 (33) | 1 (14) |
| | 3 | 1 (17) | 5 (71) |
| Primary tumor location | Right/transverse colon (%) | 4 (67) | 4 (57) |
| | Left colon (%) | 2 (33) | 3 (43) |
| Primary tumor T-stage[b] | pT1-2 (%) | 2 (33) | 0 (-) |
| | pT3 (%) | 1 (17) | 1 (14) |
| | pT4 (%) | 3 (50) | 6 (86) |
| Primary tumor N-stage[b] | pN0 (%) | 3 (50) | 0 (-) |
| | pN1 (%) | 1 (17) | 4 (57) |
| | pN2 (%) | 2 (33) | 3 (43) |
| Metastatic disease | No (%) | 4 (67) | - |
| | Yes, synchronous (%) | 2 (33) | 4 (57) |
| | Yes, metachronous (%) | - | 3 (43) |
| Peritoneal carcinomatosis index | 0 | 4 (67) | - |
| | 1–19 | 2 (33) | 4 (57) |
| | ≥20 | | 2 (29) |
| | Missing | | 1 (14) |
| Mismatch repair | Proficient – MSS (%) | 4 (67) | 5 (71) |
| | Deficient - MSI high (%) | 2 (33) | - |
| | Missing (%) | - | 2 (29) |
| Surgery | Radical primary (R0)/ complete cytoreduction | 5 (83) | 5 (71) |
| | Microscopically incomplete (R1) | 1 (17) | - |
| | Open-close | | 2 (29) |
| Chemotherapy | Yes, adjuvant | 3 (50) | 5 |
| | Yes, palliative | - | 2 |
| | No | 3 (50) | - |

[a]2 patients provided both colon and peritoneal samples.
[b]1 patient's primary tumor T- and N-stage classification was not based on pathological assessment but radiological classification.

CRC and especially metastatic MSS CRC because of its poor prognosis. While T cell infiltration, particularly of cytotoxic memory CD8[+] T cells, is a major determinant of CRC survival[10–12] and immunotherapy response[13], the role of innate lymphoid cells (ILC), including natural killer (NK) cells, in CRC and CRC-PM is less clear.

The ILC family is composed of five main subsets[14]. NK cells possess an arsenal of activating and inhibitor receptors that regulate their cytotoxic functions and secretion of cytokines and chemokines, ultimately aimed at eliminating virus-infected cells and tumor cells[15]. Related to NK cells are subsets of ILC type 1 (ILC1), which lack some prototypical NK cell features, including Eomes and perforin, but still produce interferon (IFN)-γ[16]. ILC3 and lymphoid tissue inducer (LTI) cells are, via their interleukin (IL)−22 production and formation of secondary lymphoid tissues, respectively, important for gut immune homeostasis[14]. ILC2 produces type 2 cytokines and thus contributes to parasite defense but also type 2 inflammation[17].

Tumor infiltration of immune cells expressing CD57, a marker present on certain T cells and NK cells, is linked with a favorable prognosis in CRC[18]. This is likely since CD57 marks highly differentiated, educated, and cytotoxic NK cells and T cells with high potential for anti-tumoral function[19]. However, the majority of NK cells and other ILCs in CRC tumors are CD57[−18]. Indeed, it was shown that NK cells quickly downregulate cytotoxic functions upon entry into the CRC tumor microenvironment (TME) in mice[20]. Factors such as tumor growth factor (TGF)-β, produced by cancer cells and other cells in the TME, including cancer-associated fibroblasts[21], are important in modulating the immune landscape in CRC[22]. TGF-β dampens NK cell cytotoxicity and expression of the NK cell transcription factor Eomes[23], induces tissue-resident (tr)NK cell features[23], and plays a key role in the differentiation of subsets of ILC1s[23], including intraepithelial (ie) CD103[+] ILC1[24]. Indeed, ILC1-like cells are enriched in primary CRC, along with ILC2, as compared to the unaffected colon[25]. ILC2 play ambiguous roles in animal models of CRC, showing anti-tumoral effects in some settings of CRC[26], while showing protumoral effects and being targets of checkpoint blockade in others[26,27].

Previous studies focused at addressing the role of ILCs and NK cells in primary CRC have failed to capture the large diversity of these cells in the human intestine[16] and their potential roles in CRC-PM remain unaddressed. The aim of this study was therefore to determine the ILC and NK cell landscape of primary and metastatic CRC with PM at high resolution.

Here, we use single-cell RNA sequencing (scRNAseq) to analyze healthy colon, primary CRC, and CRC-PM samples and discover that both primary CRC and PM tumors contain heterogeneous populations of ILCs and NK cells and are enriched for subsets of ILC1 and trNK cells. We also identify subsets of less differentiated intratumoral naïve (n) ILCs and early (e)NK cells with transcriptional features indicative of being precursors of more differentiated intratumoral ILCs and NK cells. Indeed, in vitro differentiation cultures reveal that intratumoral nILCs are poised for differentiation into ILC1 and NK cells with a tissue-resident phenotype, possibly contributing to their accumulation in tumors. These findings contribute to the understanding of the immune pathogenesis of CRC-PM and our large single-cell data might be used for rational design of future ILC1 and NK cell-based therapies aimed at increasing their anti-tumor activity.

## Results

### ILC heterogeneity in the unaffected colon, CRC, and PM tumors

We first assessed the infiltration of lymphocytes in the human unaffected colon (hereafter referred to as "colon"), primary CRC, and PM tumors. T cells dominated both primary CRC and, as previously reported[28], PM tumors, while ILCs and NK cells were equally frequent in tumors as compared to the colon (Supplementary Fig. 1A). ILCs and NK cells (NKG2A[−]CD127[+] ILCs, CD56[+] NK cells and CD56[−]CD7[+] non-conventional ILCs, total $n = 23407$) from 11 patients in our translational cohort contributing with tissue samples of human colon, CRC primary tumors and/or PM tumors (Table 1) were sorted by flow cytometry (Supplementary Fig. 1A, B) and analyzed by scRNAseq to assess their transcriptional heterogeneity (Fig. 1A). As the high-resolution spectrum of ILCs and NK cells in the human intestine remains underexplored, and those reports that do exist challenge previous dogmatic ILC

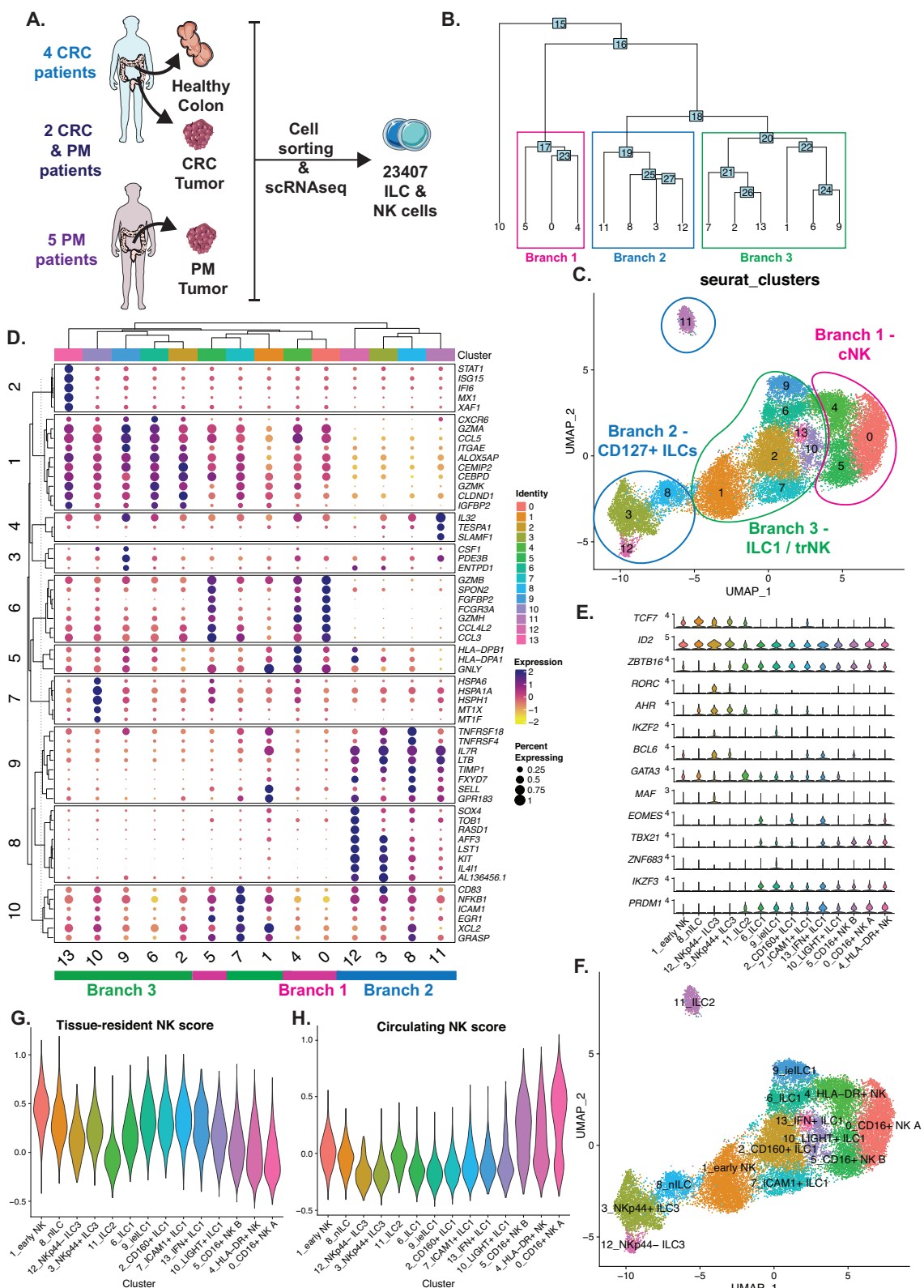

and NK cell nomenclature[16,29], we paid particular attention to anchoring our annotations in existing literature.

Hierarchical clustering of 14 transcriptionally defined clusters allowed for the identification of three main branches of clusters (Fig. 1B, C) present in the colon, primary CRC and PM tumors (Supplementary Fig. 1C, D). Differential expression (DE) analysis followed by hierarchical clustering revealed modules of genes (Fig. 1D) that

identified branch 1 as conventional (c)NK cells (module 6, including *GZMB*, *FCGR3A*, *GZMH*, and *CCL3*), branch 2 as CD127[+] ILCs (module 8, including *KIT* and *IL4I1* and module 9 including *IL7R*, *LTB* and *GPR183*) and branch 3 as ILC1/trNK cells as the clusters in this branch shared similarities with both CD127[+] ILCs and cNK cells (Fig. 1C, D). Cluster 10 was not grouped with any of the three branches in hierarchical clustering (Fig. 1B) but was included in branch 3 because of high similarity

**Fig. 1 | Characterization of ILCs in unaffected colon, primary CRC and PM tumors. A** Illustration of the cohort of patients and numbers of ILCs (NKG2A⁻ CD127⁺ ILCs, CD56⁺ NK cells and CD56⁻ CD7⁺ non-conventional ILCs) obtained after single-cell RNA-sequencing and quality control. **B** Hierarchical phylogenic tree of the 14 clusters of ILCs grouped into three branches: Branch 1 cNK (Magenta), Branch 2 CD127+ ILCs (Blue), Branch 3 ILC1/trNK (Green). **C** UMAP of the ILC single-cell dataset grouped into three branches (CD127⁺ ILC, ILC1/trNK cells, and conventional (c)NK cells). **D** Clustered dot plot showing the top five most differentially expressed genes expression and percentage of expression of each gene, allowing

identification of gene patterns. Branches are indicated under the cluster numbers (**E**). Violin Plots of selected ILC transcription factors across clusters. **F** UMAP with annotated cluster names. **G** Violin plots of trNK cell score and **H** circulating NK cell score calculated with module scoring. Data are from a total of 19 tissue samples from 11 patients analyzed in 11 independent experiments (one patient per experiment). nILC naive innate lymphoid cells, ieILC1 intraepithelial ILC1, trNK tissue resident natural killer, cNK conventional NK, CRC colorectal cancer, PM peritoneal metastasis. Illustration 1A was created using and adapted from Servier Medical Art (https://smart.servier.com), licensed under CC BY 4.0.

with other branch 3 clusters including absence of module 6, 8 and 9 genes expressed in either branch 1 or 2 clusters (Fig. 1C). More specifically, the ILC1/trNK clusters expressed genes associated with tissue residency (module 1: *CXCR6*, *ITGAE*) as well as cytotoxicity (module 1: *GZMA* and *GZMK*) while ILC1/trNK clusters 1 and 7 expressed genes in module 10 (*CD83*, *NFKB1* and *ICAM1*), also expressed by CD127⁺ ILCs. Noteworthy, clusters 1 and 8 both expressed genes of module 9, including *SELL*, typically expressed by lymphoid-homing/residing lymphocytes such as naïve T cells, nILCs, and CD56^bright NK cells[16,30,31]. Along with the DE genes (Fig. 1D), we further used a set of lineage and differentiation stage-defining transcription factors to annotate the clusters (Fig. 1E, F). Both early NK (eNK) cells (cluster 1) and naive ILCs (nILCs) (cluster 8) were defined by expression of *TCF7* and *ID2*, while ILC3 additionally expressed *RORC*, *AHR*, *BCL6* and *MAF* (Fig. 1E). ILC2 and all clusters of ILC1 and NK cells expressed *ZBTB16* (PLZF) and *PRDM1* (BLIMP-1), while ILC2 showed the highest expression of *GATA3* (Fig. 1E). *TBX21* and *IKZF3* were expressed by all clusters in the ILC1/trNK branch while *EOMES* was restricted to ILC1, CD160⁺ ILC1, IFN⁺ ILC1 as well as CD16⁺ NK cells A and HLA-DR⁺ NK cells. *ZNF683* was uniquely expressed by ieILC1 (Fig. 1E). To further align with current nomenclature, we performed a set of scorings of our clusters versus modules of recently described NK cell gene profiles. Module scoring for circulating NK cells and trNK cells[32] showed that ILC1 and trNK cells scored the highest for a trNK signature (Fig. 1G) while cNK cells scored the highest for a circulating NK cell signature (Fig. 1H). Similarly, scoring against the recently described NK1, NK2, and NK3 signatures[29] (Supplementary Fig. 1E) showed that cNK cells were most similar to NK1 cells, particularly NK1B and NK1C while HLA-DR⁺ NK cells scored highest for the NK3 signature, suggesting that they might be adaptive-like[29]. CD127⁺ ILCs and eNK cells scored the highest for the NK2 signature, while ILC1 and trNK cells showed an intermediate score for both the NK1 and NK2 signatures (Supplementary Fig. 1E). We also compared our clusters to recently published transcriptional profiles of ILC1s and NK cells in the human intestine, tonsil, lung, and peripheral blood[16] (Supplementary Fig. 1F, G), which supported our annotations of the cNK cells, CD127⁺ ILCs, ILC1 and trNK cells. In particular, we confirmed the identity of ieILC1 as *PRDM1*-expressing ILC1 (Fig. 1E) and eNK cells (cluster 1) as closely resembling CD56^bright NK cells (Supplementary Fig. 1F, G), yet with a tissue resident profile (Fig. 1G).

### Two subsets of ILCs with immature signatures in colon and tumors
Whereas nILCs have been described in the healthy and inflamed colon[31], and CD56^bright-like NK cells, sharing similarities with the eNK cells identified in this study, observed in head and neck cancer[33], their roles as progenitors to more mature ILCs and NK cells in primary and metastatic CRC remain unexplored. To define these two populations in more detail, we performed a DE analysis. Several of the top DE genes of eNK cells and nILCs were shared between the two clusters, including *IL7R*, *SELL*, *GPR183*, and *CD55* (Fig. 2A), previously described in CD56^bright NK cells[16] and nILCs and ILC progenitors (ILCp)[31,34]. DE expression of *XCL1* and *XCL2* were also shared between eNK cells and nILCs, as well as with the ILC1/trNK cell branch (Fig. 2A). However, eNK cells were clearly distinct from nILCs as they resembled CD56^bright NK cells in their expression of *IL12RB*, *CRTAM*,

*IRAK3*, *GNLY*, *COTL1*, *TNFSF9* and *KLRC1* (Fig. 2B) and scored the highest towards a SELL⁺ NK cell profile[16] (Supplementary Fig. 1F). The nILC cluster showed unique expression of *TNFRSF4* and *MFGE8* but also shared several transcripts with eNK cells (*TCF7*, *TNFRSF18*) as well as with ILC2 and ILC3, including *KIT*, *LTB*, *ICOS*, *TIMP1*, *SCN1B*, *SSBP2* and *CDKN1A* (Fig. 2C). Although nILCs shared some similarities with mature CD127⁺ ILCs, and eNK resembled CD56^bright NK cells, pseudotime analysis revealed that both clusters were located at the base of the trajectory while the most differentiated clusters ILC2, ieILC1 and HLA-DR⁺ NK ended up at the end of the predicted trajectory (Fig. 2D and Supplementary Fig. 2A), regardless from which tissue they were derived (Supplementary Fig. 2B–D). This supported the hypothesis that nILCs and eNK cells are the least differentiated of the subsets analyzed, along with NKp44⁻ ILC3, the latter showing high similarity with recently described ILC3p (Supplementary Fig. 1F). Indeed, in RNA velocity analysis projected into UMAP, both NKp44⁻ ILC3 and nILC seemed to differentiate into NKp44⁺ ILC3, while eNK cells seemed more prone to differentiate into ILC1 and trNK cells (Fig. 2E and Supplementary Fig. 2E). Velocity latent time was the highest in NKp44⁺ ILC3, ieILC1, CD16⁺ NK B, and LIGHT⁺ NK, indicating these clusters are the most differentiated (Fig. 2F).

### CRC and PM tumors are enriched for ILC1 and trNK cells
We next observed compositional changes of CD127⁺ ILCs and NK cells in tumors versus unaffected colon tissue. Whereas the NKp44⁺ and NKp44⁻ ILC3 clusters were predominantly derived from colon tissue (Fig. 3A, B), the primary tumors and metastases were the dominant sources of cells in the ieILC1, ILC1, CD16⁺ NK B, HLA-DR⁺ NK, IFN⁺ ILC1 and LIGHT⁺ ILC1 clusters (Fig. 3A). Assessing the total frequencies and patient contribution of these clusters revealed a statistically significant depletion of NKp44⁺ ILC3, and a significant enrichment of the HLA-DR⁺ NK cluster in primary tumors and metastases versus unaffected colon tissue (Fig. 3C). Differential abundance (DA) testing confirmed these differences, with DA cells in the NKp44⁺ and NKp44⁻ ILC3 clusters being more abundant in the colon, and DA cells in the cNK and ILC1/trNK branch, including HLA-DR⁺ NK but also CD16⁺ NK A, CD16⁺ NK B, ILC1, ieILC1 and LIGHT⁺ ILC1 clusters being more frequent in primary tumors and metastases (Supplementary Fig. 3A–C). While the frequencies of nILCs and eNK cells did not differ between colon, primary CRC, or CRC-PM tumors, we detected an inverse correlation between nILC frequency and a tendency for an inverse correlation between eNK cell frequency and primary CRC tumor stage (Supplementary Fig. 4A, B). We also validated the existence of our identified cell clusters, including nILC and eNK cell-like cells, in a publically available scRNA-seq dataset (GSE178341) derived from 62 primary CRC tumors (Supplementary Fig. 4C–G). Due to low cell numbers in this dataset (approximately 5600 ILC and NK cells) as compared to ours (approximately 23,400 ILC and NK cells), nILC and eNK cells were combined into one population for correlation to clinical variables. Notably, we could not validate the inverse correlation between nILC frequency and primary CRC tumor stage (Supplementary Fig. 4H), which could be due to the low cell numbers from predominantly early tumor stages in this data. Furthermore, the frequency of nILC and eNK cells was similar irrespective of MMR status, biological sex, and tumor location (Supplementary Fig. 4I–K).

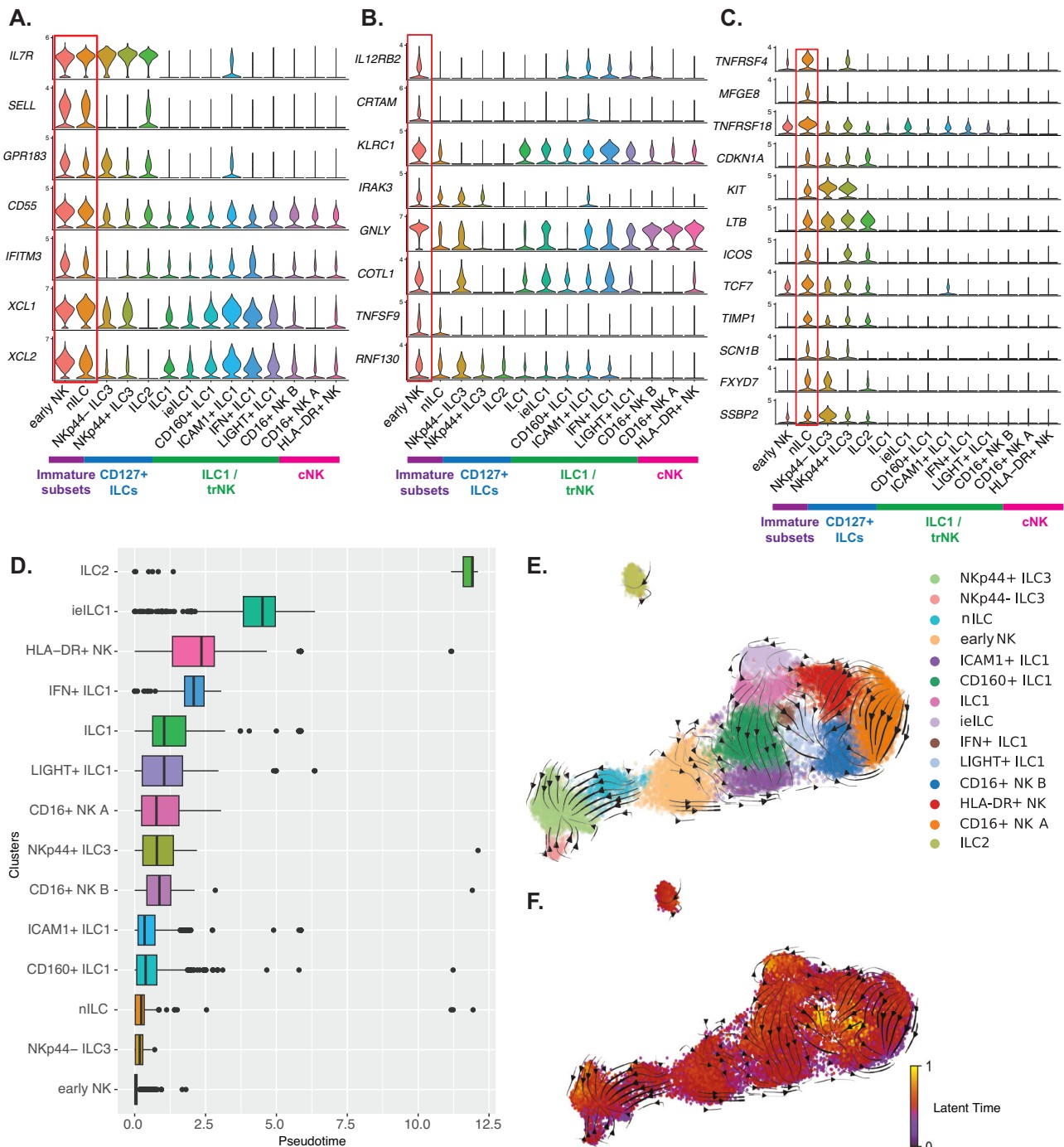

**Fig. 2 | Colon and intratumoral ILCs contain two clusters with an immature signature. A** Violin plots representing a selection of genes among the top 15 differentially expressed (DE) genes shared by the eNK cell cluster and nILC cluster. **B** DE genes expressed by eNK and **C** DE genes expressed by the nILC cluster. Cluster branches are indicated in purple for immature subsets, blue for conventional CD127⁺ ILC, green for ILC1/trNK cells and magenta for cNK cells. **D** Monocle 3 trajectory analysis and box plot representation of clusters cells across pseudotime with clusters ordered from the least to the highest mean pseudotime. Box plots show median (center line), 25th–75th percentiles (box), and whiskers extending to ±1.5× IQR; outliers plotted as individual points. Source data are provided as a Source Data file. **E** RNA velocity analysis vectors overlayed on UMAP of the single-cell dataset. **F** Velocity latent time along with velocity stream vectors, overlaid on UMAP of the single-cell dataset. Data are from a total of 19 tissue samples from 11 patients analyzed in 11 independent experiments (one patient per experiment). nILC naive innate lymphoid cells, ieILC1 intraepithelial ILC1, trNK tissue resident natural killer, cNK conventional NK.

To understand if the compositional changes in the tumors could be caused by changes in local differentiation of intratumoral progenitor cells, we performed velocity analysis on each tissue separately. This revealed a change in velocity within the eNK cell cluster in the tumors, which was the most pronounced in PM (Fig. 3D). In contrast to the unaffected colon, where eNK cell gene expression changes were directed towards the NKp44⁺ ILC3 cluster (Fig. 3D), eNK gene velocity was directed towards the LIGHT⁺ ILC1 and CD16⁺ NK B clusters in CRC primary and PM tumors (Fig. 3D).

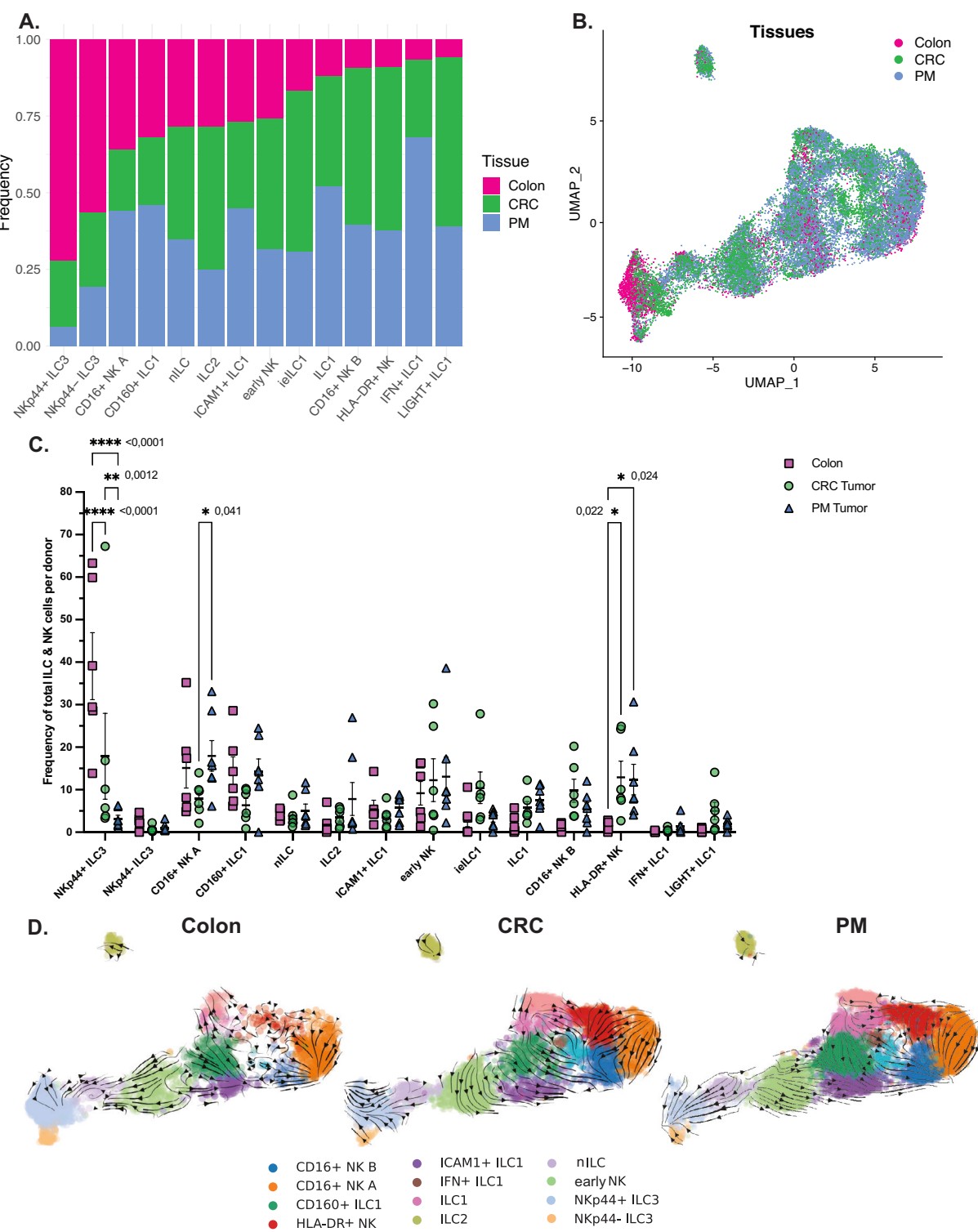

**Fig. 3 | ILC cluster compositions in unaffected colon, primary CRC and PM tumors. A** Frequency of colon, CRC, and PM tissue cells within each cluster calculated on the downsampled dataset so that each tissue contributed with the same number of total cells. **B** UMAP of the single-cell dataset representing the tissue origin of each cell. **C** Plot representing the frequency of the total cells per sample across the single-cell clusters on the downsampled dataset and split by tissue. Magenta square for colon, green circle for CRC and blue triangle for PM. Source data are provided as a Source data file. **D** RNA velocity analysis performed separately on colon, CRC, and PM cells, overlayed on a UMAP of the single-cell dataset. Data are from a total of 19 tissue samples from 11 patients analyzed in 11 independent experiments (one patient per experiment). Two-way ANOVA (two sided) with Tukey's post-hoc multiple comparisons test, data shown as mean ± SEM, adjusted *p* values * <0.05, ** <0.01, **** <0.0001. nILC naive innate lymphoid cells, ieILC1 intraepithelial ILC1, trNK tissue resident natural killer, cNK conventional NK, CRC colorectal cancer, PM peritoneal metastasis.

### nILCs and eNK cells are transcriptionally distinct in CRC tumors

We next sought to identify transcriptional differences between nILCs and eNK cells in colon versus tumors that could help explain the change in predicted differentiation trajectory in tumors identified by RNA velocity (Fig. 3D). DE analysis of nILCs and eNK cells in colon versus primary CRC tumors (Supplementary Data 1 and 2) revealed a potential ILC1 and NK-cell skewing of intratumoral nILCs, with higher expression of *GNLY, KLRC1, RGS2, CTSW,* and *CEBPB* as compared to nILCs in the colon (Fig. 4A, B). In contrast, nILCs in the colon seemed ILC3-skewed with higher expression of *LTB, PTGER4, TIPARP, IL4I1* and *IL1R1* (Fig. 4A, C), aligning with previous reports on ILC3 propensity of colon nILCs[31]. There were also transcriptional changes in intratumoral eNK cells with increased expression of *GNLY, CRTAM,* typically expressed by CD56[bright] NK cells, *SELL,* and *TCF7,* expressed by trNK cells[35], and *IL10R* and *HIF1A,* indicating influence of the tumor environment on these cells[36] (Fig. 4D, E). eNK cells in the colon expressed transcripts of early stage and trNK cells (*CD3E, ZBTB16, EOMES, IRF8*) as well as of ILC3 (*LTB, AHR*) (Fig. 4D, F), potentially explaining their predicted differentiation trajectory towards ILC3 in the colon (Fig. 3D).

### Identification and isolation of intratumoral and colonic nILC and eNK cells

To validate the existence and explore the differentiation capacity of intratumoral nILC and eNK cells, we developed a flow cytometric panel based on transcripts and selected surface protein expressions in the scRNAseq data. The nILC cluster could be distinguished from other CD127[+] ILCs by expression of *IL7R* and *KIT* but lack of *NCR2, HLA-DRA, HLA-DRB* and *PTGDR2* and from ILC1 and trNK cells by absence of *KLRD1* and *NCAM1* expression (Fig. 5A and patient characteristics in Table 2). This phenotype could be confirmed on the protein level by using the cell surface protein libraries in the single-cell dataset, identifying nILCs as expressing CD127 and CD117 but lacking CD94 and CD336 (NKp44) (Fig. 5B). The eNK cells expressed *KLRD1* and *NCAM1* but also expressed *IL7R,* distinguishing them from the ILC1 and trNK cell clusters. Furthermore, in contrast to cNK cells, they lacked expression of *FCGR3A* (Fig. 5A). On the surface protein level, the eNK cluster expressed intermediate levels of CD94 and CD127 (Fig. 5B). Additionally, both nILC and eNK cells expressed intermediate levels of the CD45RA protein (Fig. 5B).

Lin⁻CD127⁺CD94⁻ conventional ILCs, Lin⁻ CD94⁺/⁻CD56⁻CD7⁺ non-conventional ILCs and Lin⁻ CD94⁺/⁻CD56⁺ NK cells in paired colon and primary CRC tumors were analyzed by flow cytometry in a different patient group of eight patients contributing with seven colon samples and seven primary tumors (Table 2), which allowed for the identification of 17 clusters, including cells with phenotypes compatible with the eNK and nILC clusters in scRNAseq data (Fig. 5C). Cells were annotated based on unsupervised clustering followed by confirmation with manual gating for annotation (Supplementary Fig. 5). The eNK cells represented 8% of the total NK and ILC fraction (Supplementary Fig. 6) and had a profile identified as Lin⁻CD56⁺CD94⁺CD16⁻CD127^low^CD45RA⁺ (Fig. 5C–E) Notably, eNK cells were characterized by an intermediate CD127 expression, which distinguishes them from stage 4b NK cells[37] (Fig. 5D, E). The nILCs represented 7.5% of the total NK and ILC fraction (Supplementary Fig. 6) and had a profile identified as Lin⁻CD127⁺CD94⁻CD117⁺CRTH2⁻NKp44⁻HLA-DR⁻CD45RA⁺ (Fig. 5C–E). Notably, similar to our scRNAseq data (Supplementary Fig. 4A, B), we detected a tendency for an inverse correlation between nILC and eNK cell frequency and primary CRC tumor stage (Supplementary Fig. 7A, B).

### Intratumoral nILCs are prone to ILC1/trNK cell differentiation in the OP9-DL1 system

Prompted by the tumor-specific transcriptional features of nILC and eNK cells (Figs. 3D and 4), we next tested the hypothesis that intratumoral nILCs and eNK cells could have an increased capacity for ILC1 and trNK cell differentiation, thereby contributing to the

enrichment of such cells in the tumors (Fig. 3A–C). To this end, nILCs, CD127⁺ eNK cells, CD127⁻ eNK cells (defined as in Fig. 5E), NKp44⁺ ILC3 and CD16⁺ NK cells from 14 paired colon and primary CRC tumors (Table 3) were sorted by flow cytometry (Supplementary Fig. 8). Of note, due to the limited space in our flow cytometry sorting panel, CD49a⁺ cells, potentially representing CD49a⁺ stage 4b NK cells, were not excluded in the sorting of eNK cells and might represent approximately 30% of the sorted population (Fig. 4C, D). Based on the expression of cytokine receptors in the nILCs and eNK cell clusters (Supplementary Fig. 9), we co-cultured the sorted cells with OP9-DL1 cells for 14 days with either IL-2, IL-7, IL-1β, IL-12, IL-18, and TGF-β or IL-2, IL-7, IL-1β, and IL-23 (seven paired patient samples for each cytokine mix) to assess their differentiation capacity (Fig. 6A). OP9 is a mouse stromal cell line overexpressing the Notch ligand delta-like 1, known to support development of T cells[38]. However, culture of blood or tonsil ILCp with OP9, OP9-DL4 or OP9-DL1 cells and IL-2, IL-7, IL-1β and IL-23 has also been shown to support differentiation of both ILC3 and ILC1/NK cells[31,34].

Co-cultures of either nILCs, CD127⁺ or CD127⁻ eNK cells with OP9-DL1 cells and IL-2, IL-7, IL-1β, IL-12, IL-18 and TGF-β revealed extensive generation of cells expressing CD49a, CD103, NKp44, perforin and granzyme B, compatible with an ieILC1 phenotype (Supplementary Fig. 10A, B). However, we did not detect any changes in the expression of cytotoxicity-associated proteins perforin, granzyme B or K, granulysin, CD16, cytokines (IFN-γ, IL-22, and IL-13), or the tissue residency marker CD49a for neither nILC nor for any of the eNK cell cultures derived from tumor versus colon (Supplementary Fig. 10A, B). In contrast, co-cultures of nILCs with OP9-DL1 cells and IL-2, IL-7, IL-1β, and IL-23 showed that nILCs from the tumor generated more cells expressing CD117 (Supplementary Fig. 11), typically expressed by ILC3. However, analysis of the ILC3 marker CD300F and IL-22 production did not reveal any difference in the capacity of colonic nILCs to generate CD300F⁺, IL-22⁺ or IL-22-secreting ILC3-like cells (Fig. 6B). In comparison to nILCs from the colon, intratumoral nILC generated significantly more IL-13⁺ ILCs (Fig. 6C). Furthermore, intratumoral nILC generated significantly more perforin⁺ ILCs (Fig. 6D) and also seemed to generate more granzyme B expressing cells, albeit not statistically significant (Fig. 6E). Additionally, expression of the tissue residency marker CD49a showed a tendency to be higher (Fig. 6F), whereas expression of CD16 seemed to be lower in cultures of intratumoral nILCs, although not statistically significant (Fig. 6G). None of the other markers analyzed showed any clear difference following culture of intratumoral and colon nILCs (Supplementary Fig. 11).

To assess the co-expression of these molecules on nILC-derived cells following co-culture with OP9-DL1 cells and IL-2, IL-7, IL-1β, and IL-23, we turned to multidimensional analysis visualized by UMAP, which revealed 13 clusters (Fig. 6H). Cluster 0 and 2, expressing the ILC3 markers NKp44 and CD300F, of which one was expressing intermediate levels of IL-22 (cluster 2) (Fig. 6I), were enriched for cells derived from colonic nILC (Fig. 6J). However, a small cluster with high expression of IL-22, lacking NKp44 and CD300F (cluster 10), was enriched for cells derived from intratumoral nILC. Hence, the assessment of ILC3 capacity of intratumoral nILCs was inconclusive and likely unchanged.

In contrast, perforin-expressing clusters were clearly enriched in tumor-derived nILC cultures (clusters 4, 5, 9, 11; Fig. 6J). Notably, cluster 4 showed the highest expression of CD49a, with co-expression of intermediate levels of granzyme B, granzyme K and perforin (Fig. 6I). Cluster 9 showed the highest expression of CD103, with co-expression of NKp44 and intermediate levels of IFN-γ, NKp80 and perforin (Fig. 6I). Clusters 5 and 11 co-expressed IFN-γ and perforin, with or without CD49a (Fig. 6I). These data indicated an increased capacity of intratumoral nILC to generate cells expressing both cytotoxic proteins and tissue residency markers. Data are from a total of 14 tissue samples from 7 patients analyzed in 4 independent experiments

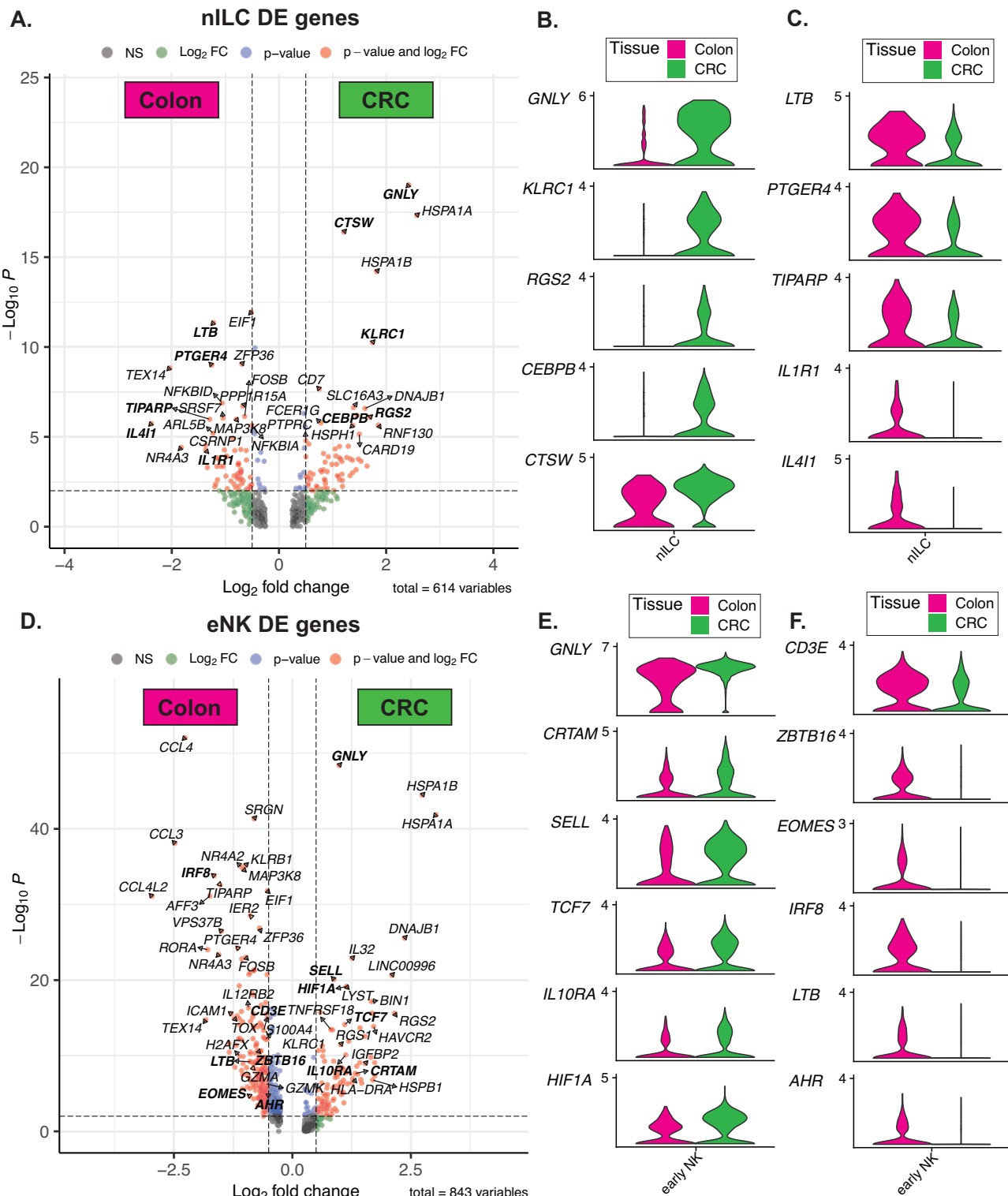

**Fig. 4 | Differential gene expression between colonic and intratumoral nILC and eNK cells. A** Volcano plot showing the differentially expressed genes between colonic (magenta) and intratumoral (green) nILC. A list of DE genes is provided as Supplementary Data 1. **B** Violin plots split by tissue of selected overexpressed genes in nILCs from CRC and (**C**) colon. **D** Volcano plot showing the differentially expressed genes between colonic (magenta) and intratumoral (green) eNK cells. List of DE genes is provided as Supplementary Data 2. **E** Violin plots split by tissue of selected overexpressed genes in eNK cells CRC and (**F**) colon. Data are from a total of 19 tissue samples from 11 patients analyzed in 11 independent experiments (one patient per experiment). Wilcoxon rank-sum test (two-sided) as implemented in Seurat, with Benjamini–Hochberg false discovery rate correction. Exact adjusted *p*-values are reported for individual genes in Supplementary Data 1 and 2. nILC naive innate lymphoid cells, CRC colorectal cancer.

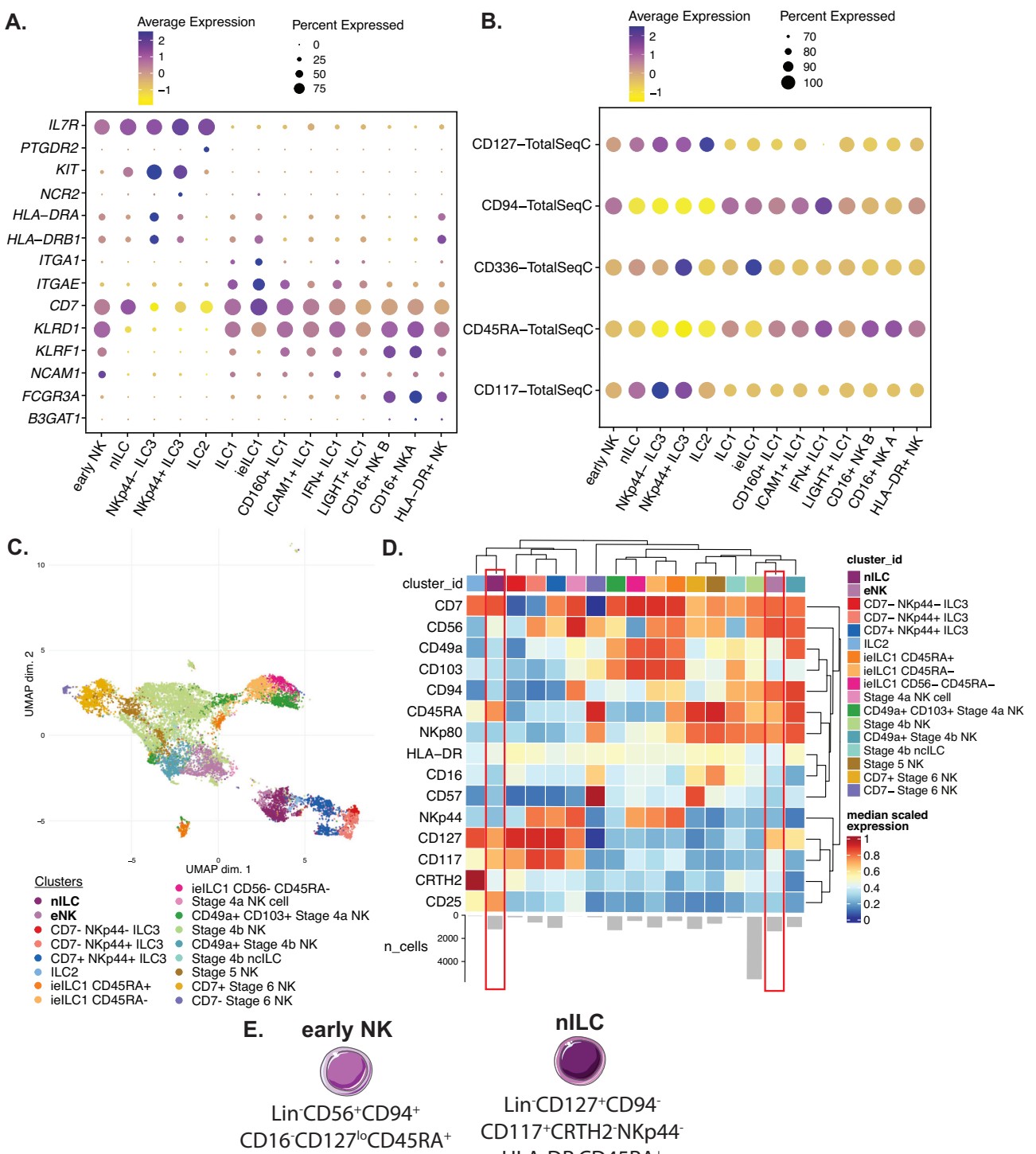

**Fig. 5 | Flow cytometric characterization of eNK cells and nILCs in unaffected colon and primary CRC tumors. A** Dot plots illustrating the gene expression and percentage of expression within the single-cell dataset of representative marker genes for eNK cell and nILC identification. **B** Dot plots of cell-surface protein expression and percentage of expression within the single-cell dataset of representative cell-surface protein markers for eNK cell and nILC identification. **C** UMAP visualization of flow cytometry data of ILCs (CD45⁺ Lineage⁻) from six colon and six primary colorectal tumors. Clusters were identified by unsupervised clustering and annotated based on their protein expression patterns and manual gating

(Supplementary Fig. 5). **D** Heatmap showing the median scaled protein expression of each ILC cluster with the bar plots representing the number or cells per cluster. Source data are provided as a Source Data file. **E** Illustration summarizing the cell-surface phenotype of nILC and eNK allowing for their distinction from other ILCs and NK cells. Data are from a total of 12 tissue samples from 7 patients analyzed in 2 independent experiments (3, 4 patients per experiment). nILC naive innate lymphoid cells, ieILC1 intraepithelial ILC1, eNK early natural killer. Figure 5E was created using and adapted from Servier Medical Art (https://smart.servier.com), licensed under CC BY 4.0.

**Table 2 | Patient characteristics of samples used in flow cytometry experiment**

| Characteristic | Subgroup | Patients (N = 8) |
|---|---|---|
| Biological sex | Female (%) | 2 (25) |
| | Male (%) | 6 (75) |
| Age at surgery | Mean years (range) | 73 (55-89) |
| Tissues donated | Normal colon and colorectal primary tumor | 5 (63) |
| | Normal colon only (%) | 1 (13) |
| | Colorectal primary tumor only (%) | 2 (25) |
| American Society of Anesthesiologists status score | 2 | 8 (100) |
| Primary tumor location[a] | Right/ transverse colon (%) | 2 (29) |
| | Left colon (%) | 5 (71) |
| Primary tumor T-stage[a] | pT1-2 (%) | 3 (43) |
| | pT3 (%) | 2 (29) |
| | pT4 (%) | 2 (29) |
| Primary tumor N-stage[a] | pN0 (%) | 4 (57) |
| | pN1 (%) | 2 (29) |
| | pN2 (%) | 1 (14) |
| Metastatic disease[a] | No (%) | 7 (100) |
| | Yes (%) | - |
| Mismatch repair[a] | Proficient – MSS (%) | 1 (14) |
| | Deficient - MSI high (%) | 1 (14) |
| | Missing (%) | 5 (71) |
| Surgery[a] | Radical primary (R0) | 7 (100) |
| Chemotherapy[a] | Yes, adjuvant | 2 (29) |
| | No | 5 (71) |

T-stage: extent of primary tumor; N-stage: regional lymph node metastasis.
[a]of patients (N = 7) that donated colorectal primary tumor.

(1–2 patients per experiment). In addition, intratumoral nILCs showed an increased capacity to generate cells expressing high levels of IL-13 (cluster 8) (Fig. 6I, J).

Together, these results indicate that while both nILCs and eNK cells can generate ILC1/trNK cells, compared to nILCs from the unaffected colon, intratumoral nILCs have an increased propensity for differentiation to cells with either an ILC1/trNK cell profile or an ILC2-like phenotype.

**Intratumoral nILCs and eNK cells generate preferentially ILC1/trNK cells in co-culture with Caco-2 cells**
To explore the differentiation capacity of nILC and eNK cells in a more tumor-relevant and NK cell-promoting setting, we co-cultured eNK cells and nILCs with the human colorectal adenocarcinoma cell line Caco-2 and IL-2, IL-7, IL-1β, IL-23, and IL-15, the latter to promote NK cell differentiation (Fig. 7A). Cells were sorted from the unaffected colon and primary CRC tumor of 10 CRC patients (Table 3). eNK cells were sorted as CD49a⁻ cells to avoid inclusion of CD49a⁺ stage 4b cells (Fig. 4C, D and Supplementary Fig. 12). Similar to co-cultures with OP9-DL1 cells, intratumoral nILCs and eNK cells generated cells expressing granzyme B (Fig. 7B), often in combination with expression of perforin, CD49a and CD103 (Fig. 7C–E and Supplementary Fig. 13A, B), indicative of ILC1/trNK cell differentiation. However, IFN-γ and Granzyme K expression was low in these cultures (Supplementary Fig. 13C, D), potentially as a result of suppressive factors such as TGF-β or other factors produced by the Caco-2 cells and/or the absence of IL-12 and IL-18. Notably, we observed high expression of NKp44 and NKG2D (Fig. 7F–I) and to a lesser extent NKp46 and NKp80 (Fig. 7J–M) on

nILCs and eNK cells after culture, giving clues as to how such ILC1/trNK cells could be activated in the TME.

Intratumoral nILCs additionally generated IL-22⁺ ILC3-like cells (Supplementary Fig. 13E). However, we observed low IL-13 expression on nILCs after co-culture (Supplementary Fig. 13F) and no divergence in the ILC1 or NK cell differentiation capacity of colon versus intratumoral nILCs (Fig. 7B, D). This contrasted with our findings with OP9-DL1 stromal cells where intratumoral nILCs generated more ILC2- as well as ILC1/NK-like cells (Fig. 6), the latter in agreement with their higher expression of NK cell genes in the tumors (Fig. 4A). It is therefore possible that the Caco-2 tumor cell microenvironment in combination with IL-15 contributed with signals that compensated for the lower intrinsic ILC1/NK cell capacity of colon nILCs while overriding the ILC2 potential of tumor nILCs. This suggests that the TME has an important role in programming the fate of nILCs.

## Discussion
CRC is the third most common form of cancer[1] and patients who develop PM have a particularly poor prognosis[39]. Most patients are not considered for surgical treatment of PM, and current chemotherapeutic treatment of CRC-PM is often inefficacious[39]. High intratumoral T cell infiltration is associated with improved survival[10–12]. Yet, immune-checkpoint blockade only benefits a small number of patients[8,9]. Hence, there is a large need for increased understanding of the immune landscape in CRC and CRC-PM, and the discovery of treatment targets.

Here, we show that primary CRC tumors and CRC-PM are infiltrated by a heterogeneous population of ILCs that includes subsets of ILC3, ILC1, trNK cells, and cNK cells. It also encompasses two immature ILC populations, one identifiable as the previously described nILCs[31,34] and the other being similar to blood CD56^bright NK cells, albeit with a tissue resident profile[16], which we refer to as eNK cells. Our study extends existing literature on the intratumoral ILC and NK cell landscape in primary CRC[40–43] to PM. Our data further aligns well with the growing literature on ILC and NK cell heterogeneity in other solid cancers, e.g., breast[43], bladder[44], head and neck[33], and lung cancer[45].

Whereas the ILC population in the unaffected colon is unsurprisingly dominated by ILC3s[40], recognized for their key role in gut homeostasis and prevention of CRC in mouse models[46], primary CRC tumors and CRC-PM were enriched for cells in the cNK branch of clusters (CD16⁺ NK A, CD16⁺ NK B, and HLA-DR⁺ NK cells). These expressed GZMB, FGFBP2, FCGR3A, GZMH, and CCL3, and showed high similarity to circulating NK cells and trajectory analysis revealed that they were transcriptionally related to each other. It is therefore feasible that these cNK cells are derived from the blood. Notably, the cNK cell clusters with the most pronounced expression of these cytotoxicity genes, CD16⁺ NK A and CD16⁺ NK B, were not significantly accumulated in tumors, possibly because of a transition to the other cNK cell cluster (HLA-DR⁺ NK), which were enriched in tumors, but with more variable expression of the cytotoxicity genes. Interestingly, the CD16⁺ NK A cell cluster was more common in PM than primary CRC tumors. Indeed, influx of NK cells into the local CRC tumor environment is well documented in mice and accompanied by rapid transcriptional and functional changes, including downregulation of molecules associated with recirculation and cytotoxicity[20]. Hence, the TME is likely shaping the heterogeneity of cNK cells that we observe in the tumors, including the differences observed between PM and primary CRC tumors, but in both tissue types, leading to an accumulation of cNK cells with a less pronounced cytotoxic transcriptome.

The tumors were further enriched for cells in the ILC1/trNK cell branch of clusters (ILC1, ieILC1, and LIGHT⁺ ILC1), lacking expression of cNK cell genes. ILC1s were similar to the previously described ieILC1[24] and trajectory analysis indeed confirmed a close relationship between these two cell populations. Fitting with this observation, ieILCs have previously shown to be enriched in primary CRC tumors[41] and in head

**Table 3 | Patient characteristics of the samples used for differentiation assays**

| Characteristic | Subgroup | Patients samples for assay with OP9-DL1 and IL-2/7/12/18/TGF-β (N = 7) | Patient samples for assay with OP9-DL1 and IL-2/7/1β/23 (N = 7) | Patient samples for assay with CACO-2 and IL-2/7/1β /23/15 (N = 10) |
|---|---|---|---|---|
| Biological sex | Female (%) | 4 (57) | 5 (71) | 5 (50) |
| | Male (%) | 3 (43) | 2 (29) | 5 (50) |
| Age at surgery | Mean years (range) | 77 (70-81) | 76 (53-90) | 72 (55-84) |
| Tissues donated | Normal colon and colorectal primary tumor | 7 (100) | 7 (100) | 10 (100) |
| Primary tumor location | Right/ transverse colon (%) | 2 (29) | 3 (43) | 4 (40) |
| | Left colon (%) | 5 (71) | 4 (57) | 6 (60) |
| American Society of Anesthesiologists status score | 1 | - | 1 (14) | 2 (20) |
| | 2 | 4 (57) | 3 (43) | 5 (50) |
| | 3 | 3 (43) | 3 (43) | 2 (20) |
| | 4 | - | - | 1 (10) |
| Primary tumor T-stage | pT3 (%) | - | 2 (29) | 4 (40) |
| | pT4 (%) | 7 (100) | 5 (71) | 6 (60) |
| Primary tumor N-stage | pN0 (%) | 3 (43) | 4 (57) | 3 (30) |
| | pN1 (%) | 2 (29) | 3 (43) | 5 (50) |
| | pN2 (%) | 2 (29) | - | 2 (20) |
| Metastatic disease | No (%) | 7 (100) | 7 (100) | 9 (90) |
| | Yes (%) | - | - | 1 (10) |
| Mismatch repair | Proficient—MSS (%) | 3 (43) | 4 (57) | 5 (50) |
| | Deficient— MSI high (%) | 2 (29) | - | 3 (30) |
| | Missing (%) | 2 (29) | 3 (43) | 2 (20) |
| Surgery | Radical primary (R0) | 7 (100) | 7 (100) | 10 (100) |
| Chemotherapy | Yes, adjuvant | 3 (43) | 3 (43) | 5[a] (50) |
| | No | 4 (57) | 4 (57) | 5 (50) |

[a]Among which two patients were recommended adjuvant chemotherapy but unknown if they received it.
T-stage: extent of primary tumor; N-stage: regional lymph node metastasis.

and neck cancer[33] where they mediate recruitment of T cells via IFN-γ-mediated induction of CXCL10[47]. LIGHT has previously been described on tumor-responsive NK cells and to mediate activation and maturation of dendritic cells[48]. Velocity analysis showed that LIGHT+ ILC1s were transcriptionally related to CD16+ mNK B but also to CD160+ ILC1, which in turn were related to eNK cells. While ieILC1 can be generated from CD56bright NK cells in blood[33], possibly as they enter tumors and the TME, our data suggest that subsets of ILC1 and trNK cells could potentially be derived from local precursors in the tumors. To address this, we sorted and cultured both nILCs, which have previously shown capacity for differentiation to Eomes+ and IFN-γ+ ILC1/NK cells in the healthy and inflamed colon[31,34] and eNK cells, with known ILC1 and NK cell differentiation capacity in tonsils and blood[33,49]. Noteworthy, the eNK cells in tumors, expressing high levels of *SELL* and *TCF7*, are similar to the recently described Tcf1high trNK cells described in a model of skin infection[35]. These Tcf1high trNK cells, which in contrast to eNK cells, express Eomes, were derived from circulating NK cells, showed features of immunological memory, and could give rise to effector cells upon rechallenge. Indeed, eNK cells generated ieILC1-like NK cells in the presence of IL-1β, IL-12, IL-18, and TGF-β, as well as with IL-1β and IL-23. Hence, similar to Tcf1high trNK cells in the mouse, eNK cells can be seen as intratumoral precursors to cytotoxic ILC1 and NK cells.

In contrast to the prediction by velocity and DE analysis, we did not detect any change in the differentiation capacity of intratumoral eNK cells. It is possible that the flow cytometric definition of nILC and eNK cells was not absolutely aligned with the transcriptional signatures, causing some eNK cells from the transcriptional definition to be sorted as nILCs. Indeed, we noticed an increased capacity for intratumoral nILCs to generate cells with an ILC1/trNK cell phenotype

in the presence of IL-β and IL-23, expressing cytotoxicity molecules such as perforin and granzyme B, while in parallel expressing tissue residency markers CD49a and CD103. These cells lacked CD16, indicating that intratumoral nILCs do not generate fully mature cNK cells. We did however, detect expression of NKp44 and NKp80 on these cells, and additionally detected NKG2D and NKp46 on ILC1/trNK cells generated in co-cultures with Caco-2 cells. Hence, our data support the concept that intratumoral ILC1/trNK cells with cytotoxic features can be generated not only from CD56bright blood NK cells, as previously reported[33], but also through local differentiation of intratumoral eNK cells and nILCs. Indeed, both Notch and IL-23, used in our OP9-DL1 differentiation system, cooperate to promote differentiation of ILC1 from nILCs in peripheral blood[50]. Noteworthy, the OP9-DL1 system additionally triggers WNT/β-catenin signaling through Notch activation, and both pathways are known to regulate NK cell and ILC3 differentiation[51]. However, while Notch signaling promotes both NK cell and ILC3 differentiation, WNT/β-catenin signaling inhibits ILC3 function[52]. For NK cells, the role of WNT/β-catenin signaling is more complex, promoting NK cell differentiation in some studies, while inhibiting NK cell differentiation in other studies[51]. Interestingly, increased WNT/β-catenin signaling has been reported in ILC3 in CRC[52]. This pathway may similarly be altered in nILCs and eNK cells. Supporting this, *TCF7*, a downstream gene of WNT/β-catenin signaling, was significantly upregulated in intratumoral eNK compared with colonic eNK, indicating WNT/β-catenin-induced modulation of eNK cells in CRC.

While OP9-DL1 co-cultures provide important insights into the differentiation capacity of intratumoral nILC, more physiologically relevant systems may be needed to capture the influence of the TME,

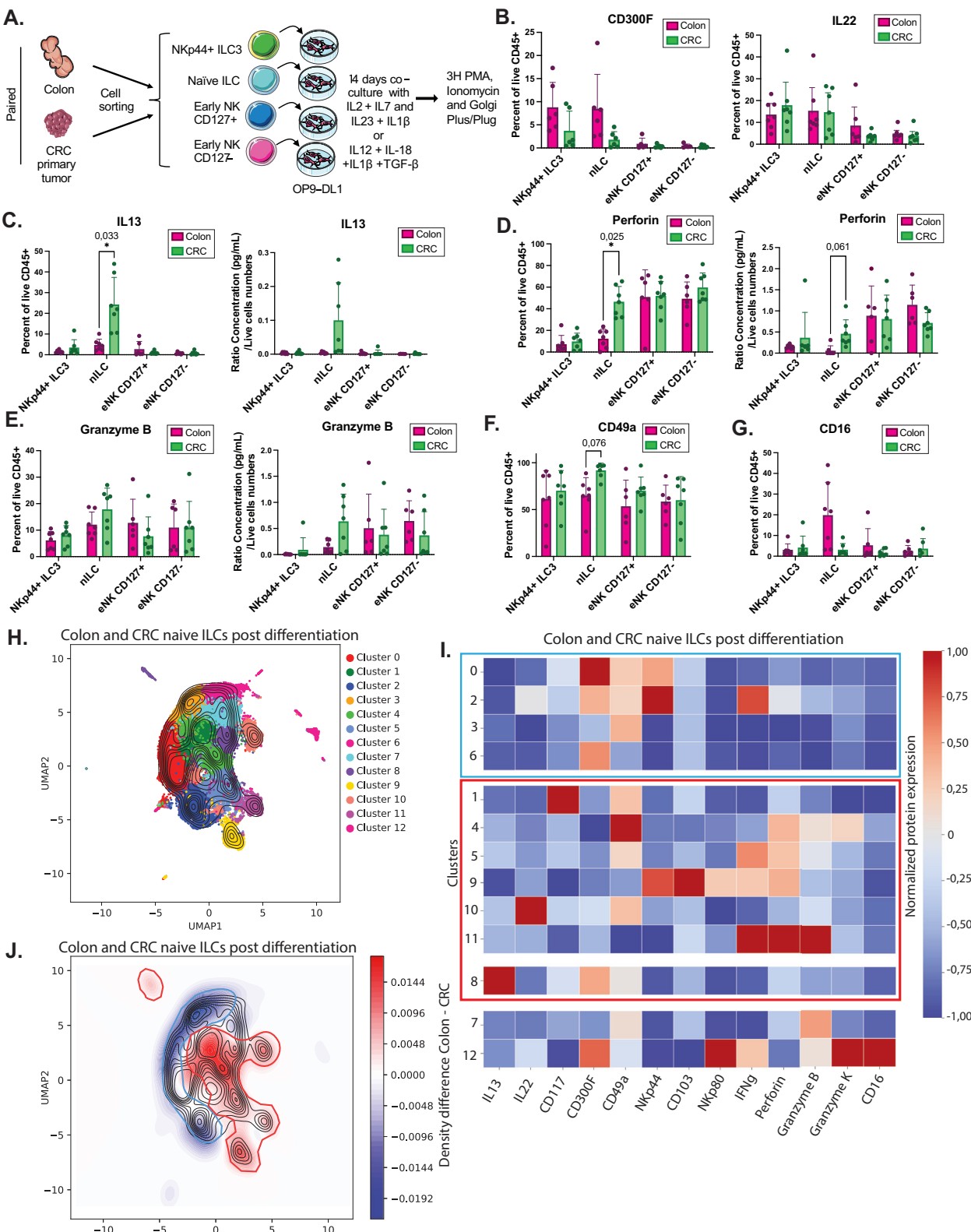

especially on NK cell differentiation. To address this, we co-cultured nILCs and eNK cells with the human CRC cell line Caco-2, along with IL-15 to promote NK cell differentiation. In this system, colon and intra-tumoral nILCs showed a similar capacity to generate ILC1/trNK cells, suggesting that the TME has a key role in shaping the differentiation capacity of nILCs. Recent advances in CRC organoid systems offer an attractive alternative platform to model ILC differentiation[53].

Furthermore, in the OP9-DL1 system, we observed an increased capacity of intratumoral nILCs to generate IL-13+ ILC2s. The ILC2-potential of intratumoral nILCs was however, not as evident in co-cultures with Caco-2 CRC tumor cells, which could be due to the NK cell skewing of this system with IL-15. We did indeed detect a small population of ILC2s in both primary CRC and CRC-PM, but observed no enrichment, irrespective of tumor stage, as compared to the

**Fig. 6 | Differentiation capacity of colon and intratumoral nILC in the OP9-DL1 system. A** Graphical illustration of the OP9-DL1 co-culture assay: Sorted nILCs, CD127+ and CD127- eNK cells, and NKp44+ ILC3 (as shown in Supplementary Fig. 8) from seven paired colon and primary colorectal tumors were co-cultured with OP9-DL1 cells for 14 days with IL-2, IL-7, IL-1β and IL-23 (data shown here in main Fig. 6) or IL-2, IL-7, IL-1β, IL-12, IL-18 and TGF-β (data shown in Supplementary Fig. 10). Supernatant collection and flow cytometry staining was performed after 3 h activation with PMA, ionomycin, Golgi Plug and Golgi Plus. **B** Bar plots representing mean ± SEM of IL-22 and CD300F, **C** IL-13, **D** perforin, **E** granzyme B, **F** CD49 and **G** CD16 protein expression measured by flow cytometry among CD45+ live cells in the different cell subsets after 14 days of OP9-DL1 cell co-culture. Calculated ratio of supernatant concentration of (**C**) IL-13 and (**D**) perforin and (**E**) granzyme B in pg/mL found by bead-based multiplex immunoassays divided by the number of live CD45+ cells acquired by flow cytometry. Colon in magenta and CRC in green. **H** UMAP showing the 13 different clusters of cells identified by flow cytometry after 14 days of co-culture of nILCs from colon and primary CRC tumors. **I** Heatmap of the mean normalized protein expression per cluster of colon and tumor nILC. **J** Cell density difference between colon (blue) and tumor (red) nILC projected on the UMAP, clusters enriched in CRC are highlighted in red and those enriched in colon highlighted in blue. Source data are provided as a Source Data file. Data are from a total of 14 tissue samples from 7 patients analyzed in 4 independent experiments (1–2 patients per experiment). Multiple paired t-test (two sided) with Holm-Šídák correction, mean ± SEM, adjusted p values * <0.05, ** <0.01, *** <0.001. nILC naive innate lymphoid cells, eNK early NK, CRC colorectal cancer, PMA phorbol 12-myristate 13-acetate. Figure 6A was created using and adapted from Servier Medical Art (https://smart.servier.com), licensed under CC BY 4.0.

unaffected colon. This aligns with our observation in the Caco-2 co-culture system. However, it contrasts with a previous study where ILC2 infiltration was negatively correlated to tumor stage[25]. Nevertheless, the increased ILC2 capacity of intratumoral nILCs in the OP9-DL1 setting is worth exploring further since ILC2s have been described to both promote[27,54] and inhibit[26] tumor growth in CRC, and to exert cytotoxic functions[55]. Of note, we observed no cytotoxic features such as granzyme B expression of intratumoral ILC2 or the intratumoral nILC-generated ILC2 in our study.

A strength of this study is the in-depth analysis of over 23,000 ILCs and NK cells in primary CRC and CRC-PM tumors using scRNA-seq, which, to our knowledge, represents the largest discrete CRC dataset on human ILCs and NK cells and the only dataset that includes non-conventional CD7+CD56-CD127- cells. It is also the first study of ILCs and NK cells in human CRC-PM, expanding our knowledge of innate immune cell infiltrates to this group of patients with poor prognosis. While the patient cohort size of this study is small and does not allow for reliable correlation with clinical parameters, it enables the study of ILC and NK cells at unprecedented granularity, which cannot be achieved by whole tumor sequencing. Our study therefore, opens an avenue for future clinical research at a wider scale. The study design included all patients planned for CRC and CRC-PM surgery, without neoadjuvant chemotherapy, at the Karolinska University Hospital, which improves the generalizability of the results and represents a patient cohort from a University hospital. However, the cohort is heterogeneous in terms of biological sex, age, tumor location, and follow-up (disease occurrence/death) and too small to reliably address the impact of these factors. Another limitation is that the comparisons between primary CRC and peritoneal metastases could have been affected by MMR status, since there are missing data on MMR status for some samples and MSI-high was more common in early-stage CRC than CRC-PM, in line with previous research[56]. MMR status is associated with NK cell infiltration, e.g., the expression of tumor checkpoints and markers of tissue residency[57]. To address some of these caveats, we made use of a publicly available scRNAseq dataset, obtained from 62 CRC tumor samples. We were able to validate the identified cell clusters but did not find any correlation between cellular frequencies and tumor stage or MMR status, which could be due to the low cell numbers in this dataset. Hence, an area of future research on a larger dataset could be to investigate if the results differ by clinical or molecular subgroups. In summary, CRC tumors studied here were biased toward more advanced stages, metastatic and MMR proficient CRC, groups that would benefit the most from therapeutic target discoveries.

Adoptive transfer of NK cells has proven successful in the treatment of several hematological malignancies[58,59]. However, there are still significant challenges with cellular therapies in solid tumors, including CRC[60]. One reason could be the lack of tumor-homing of these cellular products, which are often blood-derived or based on cell lines, e.g., NK92 or induced pluripotent stem cells (iPSC)[60]. Our results, therefore open therapeutic possibilities by modifying the local TME to enhance the local generation of intratumoral ILC1 and trNK cells with anti-tumor function. Alternatively, intratumoral nILCs and eNK cells could be harvested for the generation of potential tissue-homing ILC1 and NK cells in cases of locally advanced or metastatic CRC, which has a high risk of relapse. A recent study showed that generation of memory-like NK cells from blood NK cells in the presence of IL-12/15/18 promoted NKG2D-dependent anti-CRC tumor immunity[61]. However, it remains unclear if such cells can be derived from nILCs and eNK cells. It is also worth noting that we observed that nILCs and eNK cells are, along with ILC1 and trNK cells, expressing the chemokines *XCL1* and *XCL2*, hence potentially capable of attracting other immune cells, including CXCR1+ dendritic cells[62] and CD8+ T cells[63], to tumors. While significant research remains, our work creates a scientific foundation for the idea of harnessing the potential of intratumoral nILCs and eNK cells as progenitors of anti-tumoral ILC1 and trNK cells in CRC and CRC with PM. Indeed, our large single-cell data provide insight to be used for rational design of future ILC1 and NK cell-based therapies aimed at increasing the anti-tumor activity of these cells.

## Methods
### Participants and samples
In this translational cohort study, approved by The Swedish Ethical Review Authority to ensure it complies with all relevant ethical regulations, patients undergoing elective resection for colonic adeno-carcinomas and/or colorectal PM at the CRC unit at Karolinska University Hospital were eligible for inclusion. From April 2020, consecutive patients with primary colon cancer or PM originating from CRC were considered for inclusion. Sample collection included primary tumor, peritoneal metastases, and non-affected reference tissue, when possible. For practical reasons, only samples that could be collected before 16.00 on Mondays–Fridays and when laboratory staff were available were included in the study. Exclusion criteria were (i) neoadjuvant or adjuvant chemotherapy in the six months preceding the surgery, (ii) inability to read and consent to study information, (iii) a history of organ transplantation, and (iv) sample collection not possible based on surgeons' assessments. Reasons included risk of affecting pathological assessment of resected specimen due to, e.g., small tumors, or overgrowth of tumors, and/or practical logistical reasons. All patients gave informed written consent. Study participation did not affect the surgery itself. In case of colon tumor resection, tissue samples were obtained by a surgeon after removal of the specimen. The sampling procedure was developed together with pathologists at Karolinska University Hospital in order not to compromise that pathological evaluation. All surgeons were trained and as a reminder received written instructions prior to each case. The surgical specimen was carefully opened along the antimesenterical border from the distal or proximal end towards the tumor. If there was any macroscopical evidence of tumor overgrowth extra care was taken not to compromise the pathologist's possibility to assess the specimen. When the tumor was visible from the lumen a biopsy was taken from the tumor using either a small punch biopsy instrument or scissors.

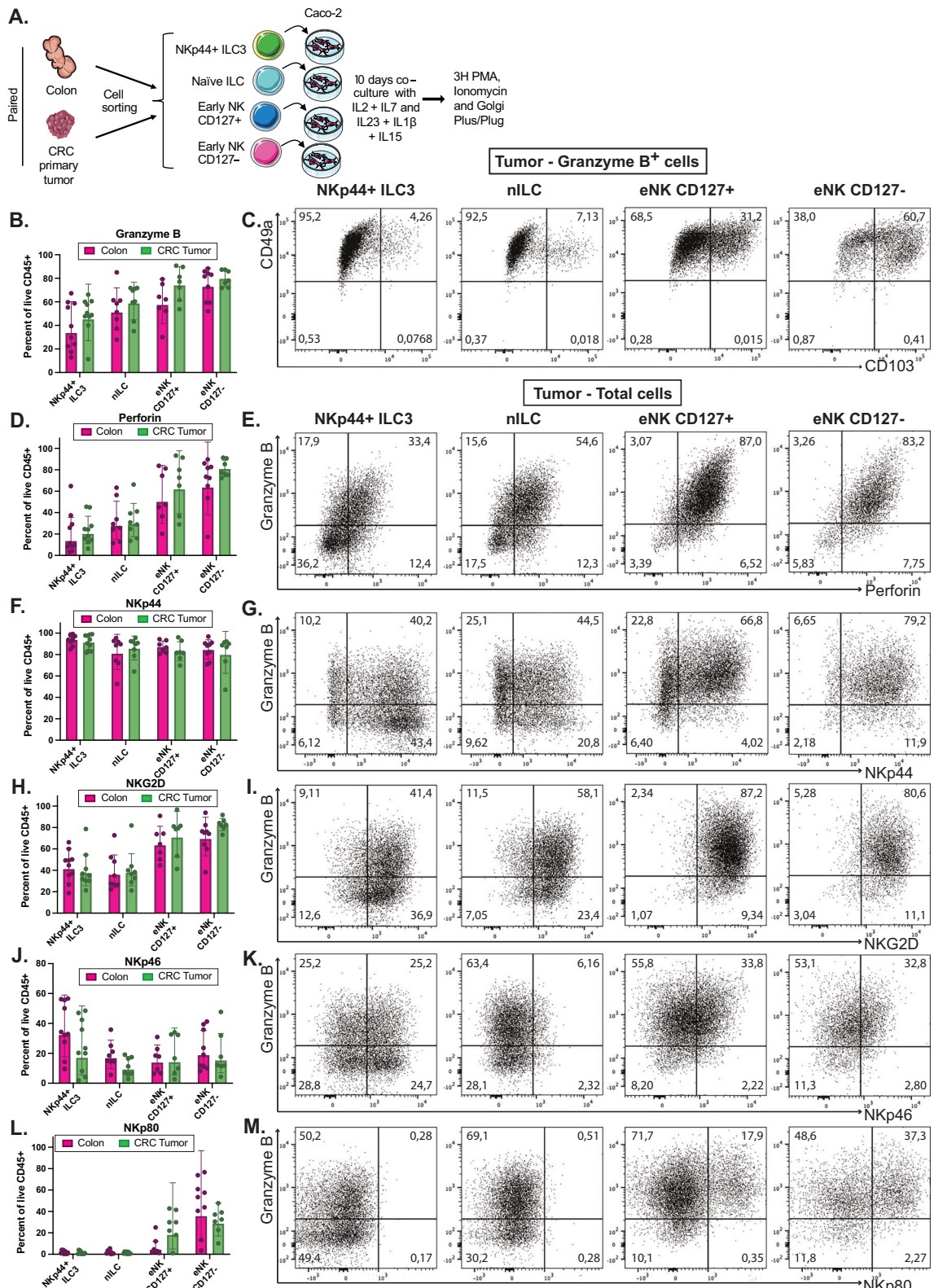

The tissue sample of unaffected colon was taken as a circular piece (approximately 1 cm) from the proximal or distal colonic marginal of the resectate, depending on the type of resection. Pathological assessment of the rest of the resected specimen followed, allowing for assessment of tumor stage reported in patient tables. Collection of data to patient characteristics tables (Tables 1–3) started prospectively and additional information regarding the pathological assessment of

the tumor and oncological treatment was collected after it became registered in the medical charts. The pathological assessment was based on the Union for International Cancer Control's Tumor-Node-Metastasis classification of malignant tumors, 8th edition[64]. For patients with PM, the tissue samples from macroscopical PM were obtained by a highly experienced consultant surgeon intraoperatively after initial assessment of the peritoneal tumor burden. Only tissue samples with

**Fig. 7 | Intratumoral nILCs and eNK cells generate preferentially ILC1/trNK cells in co-culture with Caco-2 cells. A** Graphical illustration of the Caco-2 co-culture assay: Sorted nILCs, CD127⁺ and CD127⁻ eNK cells, and NKp44⁺ ILC3 (as shown in Supplementary Fig. 12) from 10 paired colon and primary colorectal tumors were co-cultured with Caco-2 cells for 14 days with IL-2, IL-7, IL-1β, IL-23, and IL-15. Flow cytometry staining was performed after 3 h activation with PMA, ionomycin, Golgi Plug, and Golgi Plus. **B** Bar plot representing mean ± SEM of granzyme B expression. **C** FACS dotplot showing co-expression of CD49a and CD103 on granzyme B⁺ cells derived from the tumor. **D** Bar plot representing mean ± SEM of perforin expression. **E** FACS dotplot showing co-expression of granzyme B and perforin in cells derived from the tumor. **F** Bar plot representing mean ± SEM of NKp44 expression. **G** FACS dotplot showing co-expression of granzyme B and NKp44 in/on cells derived from the tumor. **H** Bar plot representing mean ± SEM of NKG2D expression.

**I** FACS dotplot showing co-expression of granzyme B and NKG2D in/on cells derived from the tumor. **J** Bar plot representing mean ± SEM of NKp46 expression. **K** FACS dotplot showing co-expression of granzyme B and NKp46 in/on cells derived from the tumor. **L** Bar plot representing mean ± SEM of NKp80 expression. **M** FACS dotplot showing co-expression of granzyme B and NKp80 in/on cells derived from the tumor. Source data are provided as a Source data file. Data are from a total of 20 tissue samples from 10 patients analyzed in 5 independent experiments (2 patients per experiment). Multiple paired t-test (two-sided) with Holm–Šídák correction, mean ± SEM, revealed no significant differences between colon (magenta) and CRC tumor (green) samples. nILC naive innate lymphoid cells, eNK early NK, CRC colorectal cancer, PMA phorbol 12-myristate 13-acetate. Figure 7A was created using and adapted from Servier Medical Art (https://smart.servier.com), licensed under CC BY 4.0.

visibly macroscopic cancerous tissue were included. The samples were taken in whole with a few millimeters of margin in the tissue. Tissue samples were placed in wash buffer containing Hanks' balanced salt solution supplemented with antibiotics as soon as removed from the patient and transported in 3–5 °C until processing. The surgery continued according to medical assessment with the planned cancer resection and/or CRS with or without HIPEC. Some patients underwent open/close procedures due to too advanced cancer burden.

Selection of samples for each experiment was based on sample availability, although a representative mix of patients (tumor stage, age, biological sex) was strived for. One patient donated samples that were used in several experimental modalities. The scRNAseq experiments required samples obtained the same day, while cryopreserved samples could be used for the other experiments.

### Cell isolation from colon, colonic primary tumors, and peritoneal metastases

Colon samples were dissected to remove fat and muscle layers. PM were dissected to isolate the tumor from the surrounding unaffected tissue, i.e., peritoneum/omentum. All samples were thinly cut into 1–2 mm pieces before subsequent enzymatic digestion into 5 ml of Iscove modified Dulbecco media (IMDM) (supplemented with antibiotics), collagenase II (0.25 mg/ml; Sigma), and deoxyribonuclease (0.25 mg/ml; Roche). Magnetic stirring was performed at 37 °C and 450 rpm for 30 min. Enzymatic digestion was stopped by adding IMDM media supplemented with 10% FBS and the samples were passed through a 40 µM cell strainer before being counted.

For biobanking, maximum 40 million cells were frozen per vial. Cells were pelleted by centrifugation and resuspended in 800 µL of ice-cold FBS and 800 µL of FBS 20% DMSO was added dropwise before cells were transferred into a cell freezing container and stored overnight at −80 °C. Cells were then transferred to liquid nitrogen for long term storage. Cells were thawed following 10X Genomics Demonstrated Protocol CG00039, Rev E, when needed for subsequent flow cytometric or in vitro differentiation analysis.

### Cell sorting

Freshly isolated cells were sorted on the same day as the surgery for single-cell sequencing and from frozen/thawed samples for the flow cytometry analysis and in vitro differentiation assays. For sorting, cells were incubated with NIR live/dead cell marker (Invitrogen) and surface protein antibodies at 4 °C for 30 min, washed with PBS and resuspended in FACS buffer (PBS with 2% FBS). Cells were sorted into 1.5 mL eppendorf tubes containing FBS on a FACSAria III sorter (BD Biosciences), with FACS Diva version 9 software.

### Single cell sequencing

Maximum 15 000 sorted CD127⁺ ILCs (CD45⁺, CD3⁻, CD19⁻, Lin⁻ and CD127⁺), NK cells (CD45⁺, CD3⁻, CD19⁻, Lin⁻, CD94⁺/⁻ and CD56⁺) and non-conventional (nc)ILCs (CD45⁺, CD3⁻, CD19⁻, Lin⁻, CD94⁺/⁻, CD56⁻ and CD7⁺) were pooled together in one 10× reaction and

15,000 T cells (CD45⁺, CD3⁺) in another 10× reaction whenever possible. The gating strategy is available in Supplementary Fig. 1A. For two samples (PM metastasis), all cells were pooled in one 10× reaction due to a small number of sorted cells. Cells were stained with TotalSeqC antibodies CD127, CD304 (NRP1), CD94, CD161, CD196 (CCR6), CD336 (NKp44), TCR Va7-2, CD294 (CRTH2), CD1d, CD45RA, and CD117 alongside surface antibody staining for cell sorting. A 5' library and a cell surface protein library were generated using the Chromium Next GEM Single Cell 5' Kit v2, Chromium Next GEM Chip K Single Cell Kit, and Library Construction Kit (10× Genomics). All steps were followed as listed in the manufacturer's instructions. Specifically, the user guide CG000331 Rev E and F.

Libraries were sequenced by SciLIfeLab (NGI; Stockholm, Sweden) using a NovaSeq 6000 sequencer and a SP-100 v1.5 flow cell (Illumina) with a read set up of 26 cycles for read 1, 10 cycles for the i7 index, 10 cycles for the i5 index, and 90 cycles for read 2.

### Analysis of single-cell sequencing data

Count matrices were generated from FASTQ files using CellRanger version 7.0.1 (10× Genomics), aligned against GRCh38, and downstream analysis was performed using the R package Seurat (version 4.3)[65–69]. Batch effects correction on donors was performed with Harmony v0.1.1[70]. For transcriptome analysis, Seurat was used for cell quality control, data normalization, data scaling, dimension reduction (both linear and non-linear), clustering, DE analysis, and data visualization. Low quality cells were removed according to the number of detectable genes (number of genes <200 or >3000 were removed) and percentage of mitochondrial genes for each cell (≥8%). Data were normalized by a log-transform function with a scaling factor of 10 000. We used variable genes in principal component analysis (PCA) and used the top 20 principal components (PCs) in non-linear dimension reduction and clustering. High-quality cells were then clustered by Louvain algorithm implemented in Seurat under the resolution of 0.6. T cells, NK cells and CD127⁺ ILCs clusters were then identified and subsetted. To isolate NK cells and CD127⁺ ILCs from T cells, cells were filtered according to their original 10x sample (NK cells and CD127⁺ ILCs only and mixed reaction with T cells, but T cells only reactions were excluded) and cells with a productive VDJ library out of the mixed reaction samples were excluded. Then, reclustering was performed with a resolution of 1.5 and further identified dead cells, doublets and non-ILC or NK cells were excluded according to the percentage of mitochondrial gene, number of features, and counts, and high CD3D, MS4A or MZB1 expression, respectively. Finally, cells were clustered again with a resolution of 0.85.

Differentially expressed genes were identified with a Wilcoxon test using Seurat function FindMarkers and defined as expressed in minimum 25% of the cluster and with a log Fold change >0.25. For nILC and eNK cells DE CRC vs. Colon analysis, only genes with a log Fold change <−0.5 or >0.5, expressed in minimum 25% of the cluster and with a p value < 0.01 were included. Volcano plots were generated using the EnhancedVolcano v1.22 package[71]. Differentially abundant

subpopulations were identified using the DASeq v1 package in R[72]. For DA testing each cell's predictive score was generated using the nearest 50 through 500 neighbors of each cell, increasing by increments of 50. Cells were then determined to be differentially abundant if they had a predictive abundance score in the top or bottom 8% of all scores, they were then divided into different subpopulations using a resolution of 0.1. Tissue composition of each cluster was performed on the dataset after subsetting each tissue to the same number of cells of the least abundant tissue (Colon with 3185 cells). Clustered dot plot was done with scCustomize v2.1.2 package with $k = 10$. Gene signatures were calculated using the AddModuleScore command from Seurat package. Velocity analysis was performed using scvelo v0.3.1 package[73,74] and plotting on the Seurat-generated UMAP. Trajectory and pseudotime analysis were done with the Monocle 3 v1.3.7 package[73,75–77].

### Differentiation cultures with OP9-DL1 cells

Stromal cell line OP9-DL1 (mycoplasma tested and passaged less than 10 times) was provided by the laboratory of Hergen Spits and was split once a week in IMDM and 10% FBS medium. ILC-OP9-DL1 co-cultures were performed in Yssel's medium supplemented with 2% normal human serum (Sigma). The day before culture, 3000 OP9-DL1 cells were seeded into a 96-well round-bottom plate to allow stromal cells to adhere. For bulk cultures, about 50–5000 sort-purified ILC and NK subsets were plated evenly on OP9-DL1 for 14 days with combinations of IL-2 (20 U/ml; PeproTech), IL-7 (PeproTech), IL-23 (R&D), IL-12 (R&D), IL-1β (R&D), IL-18 (R&D) 20 ng/ml each and TGF-β (R&D) 10 ng/mL replenishing the cytokines at day 5, 8 and 12. Half the media was replenished on days 8 and 12. At day 14, cells were stimulated with 25 ng of phorbol 12-myristate 13-acetate (PMA) and 0.5 μM ionomycin with GolgiPlug (1:10) and GolgiStop (1:15) (BD Biosciences) for 3 h before flow cytometry analysis of intracellular protein production and surface protein expression.

### Differentiation cultures with Caco-2 cells

Human colorectal adenocarcinoma cell line Caco-2 (recently authenticated, mycoplasma tested and passaged less than 10 times) was provided by the laboratory of Peter Bergman and was split once a week in IMDM and 10% FBS medium. ILC-Caco-2 co-cultures were performed in Yssel's medium supplemented with 2% normal human serum (Sigma). The day before culture, 5000 Caco-2 cells were seeded into a 96-well round-bottom plate to allow cells to adhere. For bulk cultures, about 10–5000 sort-purified ILC and NK subsets were plated evenly on Caco-2 cells for 10 days with combinations of IL-2 (20 U/ml; PeproTech), IL-7 (PeproTech), IL-23 (R&D), IL-1β (R&D) 20 ng/ml each and IL-15 (Peprotech) 10 ng/mL replenishing the cytokines at day five and day eight. Half the media was replenished on day eight. At day 10, cells were stimulated with 25 ng of phorbol 12-myristate 13-acetate (PMA) and 0.5 μM ionomycin with GolgiPlug (1:10) and GolgiStop (1:15) (BD Biosciences) for 3 h before flow cytometry analysis of intracellular protein production and surface protein expression.

### Flow cytometry

For flow cytometry analysis, cells were first stained with dead cell marker (LIVE/DEAD Fixable Near-IR, Invitrogen) and a cocktail of surface antibodies for 20–30 min at 4 °C. Antibodies reference list can be found in Supplementary Table 1. For experiments involving intracellular staining (Granzymes, granulysin, perforin and interleukines) cells were fixed and permeabilized using the BD Cytofix/Cytoperm™ Fixation/Permeabilization Solution Kit. If no intracellular staining was needed, samples were fixed for 5 min with PBS 2% paraformaldehyde buffer prior to acquisition. Samples were acquired on FACSymphony™ A3 or A5 (BD Biosciences) and analyzed on FlowJo version 10. Clustering and UMAP visualization of flow cytometry data was performed with Phenograph v1.5.2[78] and Matplotlib v3.9[79]

package on Python version 3.12 and with CiTOFWorkflow v4[80] package on R version 4.4. A detailed list of antibodies and reagents is shown in Supplementary Table 1.

### Bead-based multiplex immunoassays

Beads-based multiplex immunoassays plates for Granzyme B (analyte 72), IFN gamma (analyte 43), IL-13 (analyte 35), IL-17AF (analyte 36), IL-22 (analyte 76), and Perforin (analyte 53) were ordered from ThermoFisher Scientific. Plates were prepared according to the protocol furnished with the kit PPX-06 and acquired on a MAGPIX™ instrument. Data was analyzed with Bio-Plex Manager software.

### Analysis of publically available scRNA-sequencing data

Data from Gene Expression Omnibus series GSE178341 was reanalyzed in R using Seurat package v5.3.1. Integrated metadata annotation was used to select T cells and NK cells using "TNKILC" classification in clTopLevel. Low quality cells were removed based on the number of detectable genes (number of genes <200 or >3000 were removed) and percent of mitochondrial genes was plotted but not used to filter out cells given the high disparity between samples (from 4 up to 15%). Data were normalized by a log-transform function with a scaling factor of 10,000. We used variable genes in principal component analysis (PCA) and used the top 30 principal components (PCs) in non-linear dimension reduction and clustering. Batch effects correction on donors (PID) was performed with Harmony v1.2.4[69]. High-quality cells were then clustered by Louvain algorithm implemented in Seurat under the resolution of 0.6. T cells, NK cells and CD127+ ILCs clusters were then identified and T cells and dead cells were further removed from the Seurat object based on CD3D, TCR and mitochondrial gene expression, respectively. NK cells and CD127 + ILC clusters were subsequently reclustered with a resolution of 0.4. Label transfer was done using FindTransferAnchors() function of the Seurat package and the predicted.id added in the seurat object metadata for plotting.

### Clinical data correlations

Clinical data correlations were performed on Python v3.13.2 using the cluster (or cell population) count per sample and respective clinical data from that patient sample. Spearman rank correlations were calculated between cluster frequencies and continuous variables (tumor stage rank, age) using "scipy.stats.spearmanr" (scipy v1.16.2). Mann-Whitney U tests compared cluster frequencies across categorical variables (MMR status, tumor location, sex) using "scipy.stats.mannwhitneyu" with two-sided tests. Analyses were stratified by tissue type and cluster. Benjamini-Hochberg correction controlled for multiple testing false discovery rate. Visualizations were generated using matplotlib (v3.10.6), with scatter plots for correlations and box plots with overlaid individual points for group comparisons. Minimum sample size requirements: $n \geq 2$ per group for Mann-Whitney tests. Data processing was performed using pandas (v2.3.3) and numpy (v2.3.3).

### Statistics

GraphPad Prism 10.0 (GraphPad Software, La Jolla, CA) was used for statistical analysis. Differences between two groups were analyzed using multiple t-tests with correction for multiple comparisons. Differences among more than two groups were analyzed using two-way ANOVA followed by Tukey's multiple comparisons test with correction. Adjusted $p < 0.05$ was considered statistically significant. Parametric tests were applied as the data showed no major deviations from normality and ANOVA is robust for balanced designs with small sample sizes. Adjusted $p$ value < 0.05 was considered statistically significant. R was used for statistics of differentially expressed genes using a Wilcoxon test. Python was used to generate statistics of correlations with clinical data using a Spearman test for numerical metadata and Mann–Whitney tests for 2-group comparison.

**Reporting summary**

Further information on research design is available in the Nature Portfolio Reporting Summary linked to this article.

## Data availability

The single cell RNA sequencing data generated in this study are available in the GEO database under accession code GSE302045 (https://www.ncbi.nlm.nih.gov/geo/query/acc.cgi?acc=GSE302045). All data are included in the Supplementary Information or available from the authors, as are unique reagents used in this Article. The raw numbers for charts and graphs are available in the Source Data file whenever possible. Pelka et al.[81] scRNAseq CRC dataset is available under the accession code GSE178341 (https://www.ncbi.nlm.nih.gov/geo/query/acc.cgi?acc=GSE178341). Source data are provided with this paper.

## Code availability

All code is available in the public Github repository in the link below: https://github.com/Mjosberg-Lab/ILC-NK-HIPEC[82].

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

## Acknowledgements

We would like to acknowledge the MedH Flow Cytometry core facility (Karolinska Institutet), financed by the Infrastructure Board at Karolinska Institutet, for the use of the BD FACSAria III. We would like to thank all the surgeons at the Colorectal cancer unit at Karolinska University Hospital for their valuable contribution in collecting the tissue samples. We are grateful to Hergen Spits for the stromal cell line OP9-DL1. The authors acknowledge support from the National Genomics Infrastructure in Stockholm funded by Science for Life Laboratory, the Knut and Alice Wallenberg Foundation and the Swedish Research Council, and SNIC/Uppsala Multidisciplinary Center for Advanced Computational Science for assistance with massively parallel sequencing and access to the UPPMAX computational infrastructure. Funding was received from the Swedish Research Council (2018-00384, JM), The Swedish Cancer Foundation (19 0163 Pj 01 H, JM), The Foundation King Gustaf V Jubilee Fund (234143, CN), The Karolinska Institutet Foundations (JM), The Erling Persson Foundation (JM), the Knut and Alice Wallenberg Foundation (2014.0244 and 2018.0457, JM) and European Research Council (ERC, JM) under the European Union's Horizon 2020 research and innovation program (Grant agreement No. 850963, JM). This project was additionally supported by a donation by Mr Fredrik Lundberg (JM, CN).

## Author contributions

Conceptualization: AM, ML, CN, JM, Methodology: AM, ML, WW, CS, CAT, IM, RVP, MF, JB, LW, UL, GJP, CN, JM, Investigation: AM, ML, RVP, Visualization: AM, ML, Funding acquisition: CN, JM, Project administration: AM, ML, CN, JM, Supervision: CN, JM, Writing – original draft: AM, ML, CN, JM, Writing – review & editing: AM, ML, WW, CS, CAT, IM, RVP, MF, JB, LW, UL, GJP, CN, JM, The Colorectal Study Group contributed with methodology in terms of collection of surgical materials.

## Funding

## Competing interests

The authors declare no competing interests.

## Additional information

## Colorectal Study Group

Ali Kiasat[2], Anders Hansson Elliot[2], Carl Kördel[2], Emma Rosander[2], Henrik Iversen[2], Madelene Ahlberg[2], Mirna Abraham-Nordling[2], Petri Rantanen[2], Richard Marsk[2], Stefan Carlens[2] & Ulf O. Gustafsson[2]

