## [Transparent Peer Review file · Nature Communications]

Tumor-infiltrating immature innate lymphoid cells in colorectal cancer are biased towards tissue-resident NK cell/ILC1 differentiation

Corresponding Author: Dr Jenny Mjosberg

Version 0:

Reviewer comments:

Reviewer #1

(Remarks to the Author)

In this study, Anne Marchalot et al. analyze the spectrum of innate lymphoid cell (ILC) infiltrates in primary colorectal tumors and peritoneal metastases. They employ single-cell RNA sequencing (on 6 primary tumors and 7 peritoneal metastases), flow cytometry (on 8 primary tumors), and in vitro differentiation assays (on 7 primary tumors and 7 peritoneal metastases). Their main finding is that tumor-infiltrating immature ILCs exhibit a bias toward differentiation into tissue-resident NK cell/ILC1 phenotypes.

The study represents a detailed and compelling investigation of relatively underexplored immune cell populations. The manuscript is well-prepared, and the analyses appear technically robust. However, I have some concerns regarding the reproducibility of the results, given the limited sample size. My specific comments are as follows:

1. The small number of tumors analyzed is concerning, particularly given the known heterogeneity of immune cell infiltrates in colorectal cancer. It raises the possibility that findings may not be generalizable. The inclusion of an independent validation cohort would substantially strengthen the study. While datasets for peritoneal metastases may not be publicly available, several single-cell RNA-seq datasets exist for primary colorectal tumors. The authors might consider leveraging one of these to validate key findings. Additionally, it would be valuable, though perhaps beyond the scope of this manuscript, to assess key phenotypes in situ using multiplex immunohistochemistry or similar approaches in a larger patient cohort.
2. More detailed information about the patient cohorts is needed. When were the patients recruited? Were they a consecutive series meeting specific inclusion criteria? Do the cohorts overlap across the different experimental modalities? I recommend referring to the STROBE guidelines to enhance reporting of study participant characteristics.
3. MMR status (or microsatellite instability, MSI) is a significant confounder in colorectal cancer immunology. For instance, NK cell infiltration is typically enriched in MMR deficient tumors. It would be helpful if the authors could analyze or at least discuss the cases with "unknown" MMR status. Notably, MMR deficient tumors appear to be present among the primary tumor cases, but absent among those with peritoneal metastases, which may influence the findings and should be acknowledged in the text.
4. The biopsy procedure is currently described only briefly ("A biopsy was sampled from the primary tumor according to a protocol developed after directions from the pathology department at Karolinska University Hospital"). This description should be expanded to include key details such as method of collection, and handling.
5. Clarify how the authors ensured that the tumor and peritoneal metastasis samples were representative. This is particularly important for peritoneal lesions, where macroscopic appearance may not reliably reflect tumor presence or cellular content.
6. Given the known issues in cell culture work, including misidentification, genetic drift, and microbial contamination, the authors should describe steps taken to confirm the authenticity and integrity of the cell lines used. For example, when were the cells last tested for mycoplasma contamination? After how many passages were cells discarded or re-thawed to minimize phenotypic drift?

7. Please include catalog numbers for all reagents to improve reproducibility. RRIDs would also be useful for applicable resources such as antibodies.
8. The manuscript would benefit from including version numbers for all R packages used in the analyses to facilitate reproducibility.
9. The "Statistics" section mentions only GraphPad Prism, yet R appears to have been used for parts of the analysis. This should be clarified, and the statistical methods in R should be appropriately described.
10. In Tables I–III, T stages should be formatted as pT1–2, pT3, and pT4, rather than p1–2, p3, and p4. Similarly, N stages should be listed as pN0, pN1, and pN2.
11. Harmonize the presentation of biological sex across the tables. Table I reports percentages for both sexes, while Tables II and III only report percentages for females.
12. The discussion section would be improved by adding a dedicated paragraph acknowledging the limitations of the study, including small sample size, cohort heterogeneity, and any technical or methodological constraints.

Reviewer #2

(Remarks to the Author)

This manuscript describes the composition and heterogeneity of innate lymphoid cells in unaffected (non-tumoral) colon samples, colorectal tumors and peritoneal metastases. The authors employed scCITEseq and flow cytometry to characterize the ILC compartment leading to the identification of previously uncharacterized immature ILC subsets. They used an in-vitro system to assess the development potential of these cells showing their differentiation into ILC1/tissue resident NK cells. The novelty here lies in the characterization of the ILC infiltrate in peritoneal metastases and the identification of immature ILC populations. This is a valuable resource that presents the transcriptomic profiles of >20,000 ILCs in healthy colon and tumors which is of significant interest to the community. However, most of the results remain descriptive.

A key concern here is that the patient cohort is very heterogenous and only includes a relatively small number of samples, which somewhat limits the clinical utility of the data. Stratification analyses based on sex, primary tumor location or mismatch repair status should be performed as these factors are known to influence intratumor immune cell composition and function.

The developmental potential of nILC and eNK subsets need be further investigated, including using in vivo models and analyzing the generated intestinal ILC subsets. The use of NSG mice that express human cytokines or the use of NSG mice injected with human cytokines would represent a more physiological system to understand the developmental potential of these immature ILC subsets in the intestine. Based on the gating strategy used to cell sort eNK cells (Fig S5), it is possible that this population was contaminated by CD49a+ stage 4b NK cells. These cells express all the eNK cell markers and therefore could not be discriminated by the sorting strategy employed by the authors, unless CD49a expression was used to exclude them. Could the authors comment on their gating strategy and the purity of their eNK cell subset for their in vitro assays.

Finally, as mentioned by the authors, ILCs represent yet untapped therapeutic targets and as such, the authors should extend their flow cytometric analyses and map intratumoral ILC infiltration to correlate the proportion of eNK, nILCs and or other identified ILC subsets with disease prognosis, including the likelihood of developing peritoneal metastases. Such analyses would significantly enhance the scope of this manuscript and highlight the clinical importance of these ILC populations in colorectal cancer.

Reviewer #3

(Remarks to the Author)

This paper provides new insights into the populations of ILCs and NK cells present in CRC-PM by utilising single cell sequencing and flow cytometry. The authors identified putative precursor ILC populations that may be of interest for future research endeavours. This work is original and provides value to the field.

Below are some points to be considered.

1. The statistical tests used in figure 3 and figure 6 are not referenced in the figure legends, nor is a legend for p-value indicators – this information should be included. The methods indicate that only t-tests have been used for statistical comparisons. Particularly when multiple groups are being compared, an omnibus test should be used followed by a posthoc test with adjusted p-values so that significance is not overestimated.
2. An effort should be made to improve the interpretability of figure 1, particularly around the numbering and naming of the clusters. For example, in 1D, the key should be titled 'cluster' instead of 'identity'. A table or otherwise should be included to correlated annotated clusters with original cluster numbers rather than having to infer between UMAP plots. Additionally, cluster 10 appears to have been included as part of Branch 3 despite not clustering as such in 1B.
3. The axis labels in 2D should be cleaned up.
4. It may be useful to include the latent time plots alongside your existing RNA velocity cluster plots to improve interpretability of start and endpoints.

5. Figure S3 should have 'differential' rather than 'diferencial' in the titles.
6. You need to include information about the clustering and annotation method for the flow data in figure 5 e.g. unsupervised clustering followed by confirmation with manual gating for annotation.
7. You should include the gating strategy used for the flow cytometry in figure 5 as a supplementary figure.
8. Line 241-244 – be more clear about what comparison you are trying to make here.
9. In figure S7B you should include the concentration in your concentration/cell CBA plots. I am not sure it is correct to display your concentrations per cell here when you only know the final cell numbers but haven't measured the evolution of these populations over time. This is also true for your CBA plots in figure 6.
10. Line 244-246 – There is no significant difference here, be careful not to bias your interpretation.
11. Line 253 – there is a significant difference indicated for CD117.
12. You should include discussion around (or mention of) organoids as an alternative differentiation platform for ILCs e.g. <https://doi.org/10.1016/j.celrep.2022.111281>. Could also discuss WNT signalling as it may affect ILC differentiation in CRC tumours.

Version 1:

Reviewer comments:

Reviewer #1

(Remarks to the Author)

I think the authors have appropriately addressed the comments.

Reviewer #2

(Remarks to the Author)

I would like to thank the authors for their thorough response to my comments. They have addressed all my comments satisfactorily, and I am confident that this manuscript will serve as a strong reference in the field of ILCs in CRC.

I have a few additional minor comments regarding the newly added data.

1. Lines 259-260. Purity of the sorted eNK CD127+ cells. Could the authors indicate in the figure legends that CD49+ stage 4B NK cells represent approximately ~30% [range:0-70%] of eNK cells. This would help readers better interpret the data shown in Figure 6 and Figure S10.
2. Line, 319, the authors suggest that TGF- β production by Caco-2 cells impairs the expression of IFN- γ and granzyme K. However, in the OP9-DL1 system, the culture of nILC or eNK cells (+/-stage 4B NK cells) in the presence of TGF- β results in the accumulation of IFN- γ and granzyme K in the supernatant (Fig. S10). Could the authors comment on these potential discrepancies? Given the potential uncertainties regarding the involvement of TGF- β in this phenotype, I recommend softening this claim and possibly removing TGF- β from the sentence.
3. Line 320, the figure number is missing; it should read (Fig. 7F-I).
4. In Fig 7C, E, G, I, K, M, could the authors include representative FACS plots for NKp44+ ILC3s.
5. Line 386. There is a typo – it should read IL-1 β .

Reviewer #3

(Remarks to the Author)

The revised manuscript and response to reviewers by Mjosberg et al have largely covered the concerns and suggestions I had with one minor point remaining.

1. Regarding the updated statistical analysis, you are now utilising two-way ANOVA but this is not actually indicated in the text. You should include this and also justification for using a parametric test on a relatively small dataset.

Authors' response to the reviewer's Comments

Reviewer #1 (Remarks to the Author)

In this study, Anne Marchalot et al. analyze the spectrum of innate lymphoid cell (ILC) infiltrates in primary colorectal tumors and peritoneal metastases. They employ single-cell RNA sequencing (on 6 primary tumors and 7 peritoneal metastases), flow cytometry (on 8 primary tumors), and in vitro differentiation assays (on 7 primary tumors and 7 peritoneal metastases). Their main finding is that tumor-infiltrating immature ILCs exhibit a bias toward differentiation into tissue-resident NK cell/ILC1 phenotypes.

The study represents a detailed and compelling investigation of relatively underexplored immune cell populations. The manuscript is well-prepared, and the analyses appear technically robust. However, I have some concerns regarding the reproducibility of the results, given the limited sample size. My specific comments are as follows:

1. The small number of tumors analyzed is concerning, particularly given the known heterogeneity of immune cell infiltrates in colorectal cancer. It raises the possibility that findings may not be generalizable. The inclusion of an independent validation cohort would substantially strengthen the study. While datasets for peritoneal metastases may not be publicly available, several single-cell RNA-seq datasets exist for primary colorectal tumors. The authors might consider leveraging one of these to validate key findings. Additionally, it would be valuable, though perhaps beyond the scope of this manuscript, to assess key phenotypes in situ using multiplex immunohistochemistry or similar approaches in a larger patient cohort.

Reply: We thank the reviewer for this comment and agree that our results need to be followed up in a larger patient cohort in the future. We have highlighted this in the section related to strengths and limitations of our study (pages 14-15, lines 437-463).

Our study however represents the most detailed analysis of the ILC-NK cell spectrum of primary CRC to date, and the first ever in CRC-PM tumors. Our scRNAseq dataset contains 23,407 CD127⁺ ILCs, CD56⁺ NK cells as well as Lin⁻CD7⁺CD56⁻CD127⁻ non-conventional ILCs, which, to our knowledge, no previously generated single-cell RNA-sequencing (scRNAseq) data can be compared to. In addition to the 11 patients included for scRNAseq analysis, we included 8 more CRC patients in our flow cytometry validation experiments, and another 24 CRC patients in our functional validation experiments. Hence, data from a total of 43 patients, obtained by four different experimental methods, univocally showed the existence of immature subsets of ILCs and NK cells in CRC tumors. We are therefore confident that the main message of our paper, which is that intratumoral tissue-resident immature ILCs and NK cells can generate differentiated subsets of ILC1 and trNK cells in CRC tumors, stands strong.

Still, the reviewer's point is well taken, and we have performed an extensive inventory of previous scRNAseq studies of primary CRC to identify one or several datasets to use as an independent validation cohort (Reviewer Table I). Unfortunately, many studies do not make their data publicly accessible. We have for several years tried to get in touch with the authors of Qi et al, Cell Rep Med, 2021 (PMID 34467243) who have deposited their

data in a Chinese repository with restricted access. This data contains 31,246 CD127⁺ ILCs (but no NK cells) from 4 patients (CRC tumor+adjacent colon), which could have been useful for validation of our findings related to CD127⁺ ILCs. Unfortunately, our multiple data requests have not been answered.

Another challenge is that most previous studies include few patients (max 10, i.e. smaller than our dataset) and analyses total single cell suspensions from tumors. Since ILCs and NK cells make up only 0.1-10% of total CD45⁺ lymphocytes (Fig S1A), which in turn represent maximally 10% of total viable cells in the tumors (Fig S1A), a scRNAseq dataset of e.g. 200 000 total cells, which is considered large, would contain only 20-2000 ILC/NK cells. This shows the strength of the approach that we have used, i.e. pre-enriching for the target populations of interest. This makes our dataset unique in granularity, i.e. making it possible to discover novel ILC/NK cell states. Our inventory did however reveal one study with large numbers of patients (n=62) and decent amounts of cells (Reviewer Table I; Pelka et al: GSE178341, yellow-marked)

Reviewer Table I. Available scRNAseq data from selected CRC studies.

Accession no	No of primary CRC samples	No of total CD45+ and CD45-cells
GSE221575	9	39,484
GSE178341	62	127,695
GSE146771	18	43,817
GSE178318	6	111,292
GSE225857	4	196,473
GSE232525	Not stated	7,343
GSE242271	5	24,111
GSE200997	16	26,173

We reanalyzed this dataset (Reviewer Fig. 1) and could conclude that the number of total ILC and NK cells per sample and cell quality was low compared to our dataset, despite filtering of low-quality cells (Reviewer Fig. 1 A-B). This shows the strength of our scRNAseq dataset that although derived from 11 patients, the cell counts and data quality is high. Nevertheless, we performed cell label transfer analysis. We could indeed identify cells in the Pelka et al data that matched with the clusters that we revealed in our own dataset (Reviewer Fig. 1C) but the cell numbers were low (Reviewer Fig. 1D-E), making correlations to clinical data challenging.

Reviewer Figure 1. A. Total ILC and NK cells count per sample found in Pelka et al. (GSE178341) and our data (Marchalot et al). B. Mean count RNA per sample. C-D. Label transfer displayed on UMAP. E. Cell counts per cluster.

To understand the distribution of eNK cells and nILCs across clinical and molecular phenotypes we therefore combined these cells into one “immature subsets” (reviewer Fig. 1D) and could see that the majority of the tumors in this dataset had equal frequencies of such cells irrespective of MMR status, biological sex, tumor location and tumor stage (Reviewer Fig. 2). Of note, of the 62 patients in the dataset, 28 had clinical metadata and were included in the analysis.

Reviewer Figure 2. A-C: Total nILC and eNK cells per tumor sample combined as a frequency of “immature subsets” in A. MMR proficiency (p) or MMR deficient (d) tumors, B. Males (M) or females (F) and in tumors from right or left colon. D: Frequency of “immature subsets” (nILCs plus eNK cells) across tumor stage (T). All scRNAseq data from Pelka et al. (GSE178341) using cell labels from our scRNAseq data.

We have now included part of these data in Figure S4 and they are mentioned in the text on page 7, lines 188-196.

Finally, the reviewer suggested that we substantiate our findings with multiplex immunohistochemistry or similar approaches in a larger patient cohort. While we fully agree that this would be valuable, we do consider, which the reviewer also indicates, that this is beyond the scope of this manuscript.

2. More detailed information about the patient cohorts is needed. When were the patients recruited? Were they a consecutive series meeting specific inclusion criteria? Do the cohorts overlap across the different experimental modalities? I recommend

referring to the STROBE guidelines to enhance reporting of study participant characteristics.

Reply: Thank you for this comment which helped us improve the clarity regarding the patient cohorts in the manuscript. We have followed the STROBE guidelines to improve the clarity in study patient characteristics and study design including how patients were recruited, how samples were obtained and for what analysis. Due to low cell numbers obtained from these small tissue pieces, only one patient had enough sample to be used for several experimental modalities (scRNAseq and differentiation assay with IL-2/7/12/18/TGF-b).

The changes related to this issue can be found on page 4, 98-99; page 8, 239-240; page 9, 263-264; page 16, 487-528.

3. MMR status (or microsatellite instability, MSI) is a significant confounder in colorectal cancer immunology. For instance, NK cell infiltration is typically enriched in MMR deficient tumors. It would be helpful if the authors could analyze or at least discuss the cases with “unknown” MMR status. Notably, MMR deficient tumors appear to be present among the primary tumor cases, but absent among those with peritoneal metastases, which may influence the findings and should be acknowledged in the text.

Reply: We thank the reviewer for their comment and for highlighting the association between MMR deficiency, tumor stage and NK cell infiltration. We have reported all available data on MMR and have been able to access additional information on two patients that had unknown MMR status (see patient tables I-III). Unfortunately, not all CRC cases have been tested for MMR status at our hospital during the study period. As the reviewer highlights, MMR deficiency is more common in early-stage CRC than in metastatic disease. While our results mirror real world data on MMR status by stage, the differences in MMR status between groups could have affected comparisons between primary tumor and metastasis. This has been added to the discussion pages 14-15, lines 437-463.

Inspired by the reviewer’s question we compared MSS status in our scRNAseq data. We detected two interesting differences where the data was non-overlapping for the two groups (shown below for the reviewer’s inspection only). However, given the low frequency of MSI cases, we cannot draw any conclusions from this data and the cell numbers for these two clusters in Pelka et al (GSE178341) were too small to validate these findings (Reviewer Fig. 1E).

Reviewer Figure 3. Frequencies of nILCs and IFN+ ILC1 in MSS and MSI primary tumors.

4. The biopsy procedure is currently described only briefly ("A biopsy was sampled from the primary tumor according to a protocol developed after directions from the pathology department at Karolinska University Hospital"). This description should be expanded to include key details such as method of collection, and handling.

Reply: Thank you for this suggestion. We have expanded on this (page 17, lines 508-520).

5. Clarify how the authors ensured that the tumor and peritoneal metastasis samples were representative. This is particularly important for peritoneal lesions, where macroscopic appearance may not reliably reflect tumor presence or cellular content.

Reply: Thank you for this suggestion. All sampling was conducted by highly experienced surgeons, each with extensive expertise in CRC surgeries, including HIPEC. As the reviewer is probably aware, pathological reviews are often performed during surgeries when there is uncertainty about the presence of carcinomatosis. However, in these cases, no formal pathological review was carried out since only tissue samples with visibly macroscopic cancerous tissue were included. This is clarified in the manuscript page 17, lines 514-517.

6. Given the known issues in cell culture work, including misidentification, genetic drift, and microbial contamination, the authors should describe steps taken to confirm the authenticity and integrity of the cell lines used. For example, when were the cells last tested for mycoplasma contamination? After how many passages were cells discarded or re-thawed to minimize phenotypic drift?

Reply: OP9-DL1 cells were tested for mycoplasma once every month and were always mycoplasma negative. They were discarded at the latest after 10 passages and cells from re-thawing were passaged only once before freezing down a new stock of cells. For the Caco-2 cell line (new figure 7) we were able to confirm its authenticity and integrity (see report below). We have also added this information in the Reporting summary. However, for the OP9 mouse stromal cell line, it requires a separate kit from ATCC for authentication which unfortunately is backordered until after the deadline of this revision. We were therefore unable to perform this analysis. However, we have used this cell line since 2016 (max 10 passages) and published results (Kokkinou et al, Sci Imm, 2022) in line with others' results from this cell line (Nagasawa et al, J Exp Med, 2019),

strengthening the authenticity of the OP9 cell line. We have not used any other cell lines in parallel in our lab, making mix-ups unlikely.

**Cell Line
Authentication Service
STR Profile Report**

FTA Barcode: STRD3674
ATCC Sales Order: SO5967093

Test Results for Submitted Sample				ATCC Reference Database Profile			
Locus	Query Profile: Caco-2			Database Profile: Caco-2; Colon Adenocarcinoma; Human (Homo sapiens)			
D3S1358	14	17		14	17		
TH01	6			6			
D21S11	30	32		30	32		
D18S51	12			12			
Penta_E	7						
D5S818	12	13		12	13		
D13S317	11	13	14	11	13	14	
D7S820	11	12		11	12		
D16S539	12	13		12	13		
CSF1PO	11			11			
Penta_D	9	11					
Amelogenin	X			X			
vWA	16	18		16	18		
D8S1179	12	14		12	14		
TPOX	9	11		9	11		
FGA	19			19			
D19S433	15						
D2S1338	17	25					
Number of shared alleles between query sample and database profile:							23
Total number of alleles in the database profile:							23
Percent match between the submitted sample and the database profile:							100
The allele match algorithm compares the original CODIS 13 loci only, even though alleles from all loci and amelogenin will be reported when available.							
NOTE: The original CODIS 13 loci and amelogenin may be made public to verify cell identity. In order to protect the identity of the donor, please do not publish the allele calls from all the STR loci tested, appearing in this table with grayed fields. Electropherograms showing raw data are attached.							

7. Please include catalog numbers for all reagents to improve reproducibility. RRIDs would also be useful for applicable resources such as antibodies.

Reply: We thank the reviewer for highlighting this and we have now included catalogue numbers or RRIDs for all reagents. Changes can be found in Table S3.

8. The manuscript would benefit from including version numbers for all R packages used in the analyses to facilitate reproducibility.

Reply: We agree with reviewer that this is good practice. We have included version numbers for the R packages used. Changes can be found on after each package mention in the methods section (page 19, lines 596, 597, 604, 606 and 607).

9. The "Statistics" section mentions only GraphPad Prism, yet R appears to have been used for parts of the analysis. This should be clarified, and the statistical methods in R should be appropriately described.

Reply: While R has been used for data exploration and visualization, only GraphPad Prism was used for statistical analyses except for differentially expressed genes

statistics and new correlations statistics asked for during revisions. A clarification has been added page 22, line 686-690.

10. In Tables I–III, T stages should be formatted as pT1–2, pT3, and pT4, rather than p1–2, p3, and p4. Similarly, N stages should be listed as pN0, pN1, and pN2.

Reply: Thank you for this suggestion. We have updated tables I-III accordingly.

11. Harmonize the presentation of biological sex across the tables. Table I reports percentages for both sexes, while Tables II and III only report percentages for females.

Reply: Thank you for noticing this inconsistency. Tables I-III have been updated.

12. The discussion section would be improved by adding a dedicated paragraph acknowledging the limitations of the study, including small sample size, cohort heterogeneity, and any technical or methodological constraints.

Reply: We agree with the reviewer and have added a section “*Strengths and limitations of the study*” to the discussion on pages 14-15, lines 437-463.

Reviewer #2 (Remarks to the Author)

This manuscript describes the composition and heterogeneity of innate lymphoid cells in unaffected (non-tumoral) colon samples, colorectal tumors and peritoneal metastases. The authors employed scCITEseq and flow cytometry to characterize the ILC compartment leading to the identification of previously uncharacterized immature ILC subsets. They used an in-vitro system to assess the development potential of these cells showing their differentiation into ILC1/tissue resident NK cells. The novelty here lies in the characterization of the ILC infiltrate in peritoneal metastases and the identification of immature ILC populations. This is a valuable resource that presents the transcriptomic profiles of >20,000 ILCs in healthy colon and tumors which is of significant interest to the community. However, most of the results remain descriptive.

1. A key concern here is that the patient cohort is very heterogenous and only includes a relatively small number of samples, which somewhat limits the clinical utility of the data. Stratification analyses based on sex, primary tumor location or mismatch repair status should be performed as these factors are known to influence intratumor immune cell composition and function.

Reply: We thank the reviewer for this suggestion. As the reviewer rightly points out, our study aimed at performing the first in-depth analysis of the intratumoral ILC/NK cell compartment in a limited sample cohort, rather than a shallower analysis across more patients. While our strategy is warranted, we agree that stratification across biological sex, primary tumor location and MMR status would significantly strengthen our study. Although our study is too small to statistically verify differences in cell frequencies depending on clinical variables, we did identify some interesting cell types where the cell frequencies were non-overlapping between groups. Two such differences are seen below (Reviewer Fig. 4). The frequencies of nILCs and IFN+ ILC1 were higher in the two MSI cases as compared to the four MSS cases but we have refrained from adding it to

the paper due to lack of statistical power. We identified no such differences for sex or tumor location, which could be due to a lack of statistical power in our cohort.

Reviewer Figure 4. Frequencies of nILCs and IFN+ ILC1 in MSS and MSI primary tumors.

We additionally performed correlation analyses between cell cluster frequencies and tumor stage and identified that the frequencies of nILCs and eNK cells were inversely correlated to tumor stage. This was seen in both the scRNAseq data and for the flow cytometry data. These data have been added to Fig. S4A-B and Fig. S7A-B and are being mentioned on page 7, lines 186-188 and page 9, lines 249-250.

Still, our data were derived from relatively few patients. So, to potentially better answer the reviewer's question, we also made use of a publicly available scRNAseq dataset from 62 primary CRC tumors from an equal number of patients (Pelka et al, GSE178341). We reanalyzed this dataset (Reviewer Fig. 5) and could conclude that the number of total ILC and NK cells per sample and cell quality was low compared to our dataset, despite filtering of low-quality cells (Reviewer Fig. 5A-B). The total ILC+NK cell number in this dataset was 5631 cells (of which 4369 cells had associated clinical metadata), to be compared to our dataset of 23 400 ILC+NK cells.

This shows the strength of our scRNAseq dataset that although derived from 11 patients, the cell counts and data quality is high. Nevertheless, we performed cell label transfer analysis. We could indeed identify cells in the Pelka et al data that matched with the clusters that we revealed in our own dataset (Reviewer Fig. 5C), but the cell numbers were low (Reviewer Fig. 5D), making correlations to clinical data challenging.

Reviewer Figure 5. A. Total ILC and NK cells count per sample found in Pelka et al. (GSE178341) and our data (Marchalot et al). B. Mean count RNA per sample. C-D. Label transfer displayed on UMAP. E. Cell counts per cluster.

Due to the low numbers of eNK cells and nILCs, but to understand distribution of eNK cells and nILCs across clinical and molecular phenotypes, we combined these cells into one “immature subsets”. Of note, of the 62 patients in the dataset, 28 had clinical metadata. Using this data, we could see that the majority of the tumors in this dataset had equal frequencies of “immature subsets” (nILC+eNK cells) irrespective of MMR status, biological sex, tumor location or stage (Reviewer Fig. 6A-D).

Reviewer Figure 6. A-C: Total nILC and eNK cells per tumor sample combined as a frequency of “immature subsets” in A. MMR proficient (p) or MMR deficient (d) tumors, B. Males (M) or females (F) and in tumors from right or left colon. D: Frequency of “immature subsets” (nILCs plus eNK cells) across tumor stage (T). All scRNAseq data from Pelka et al. (GSE178341) using cell labels from our scRNAseq data.

We have added parts of these data to Fig S4C-G and included text regarding this on page 7, lines 188-196.

Finally, while our study was limited in terms of samples, it provides a basis for future research. This is now discussed more explicitly in the manuscript (discussion - *Strengths and limitations*) on pages 14-15, lines 437-463.

2. The developmental potential of nILC and eNK subsets need be further investigated, including using in vivo models and analyzing the generated intestinal ILC subsets. The use of NSG mice that express human cytokines or the use of NSG mice injected with human cytokines would represent a more physiological system to understand the developmental potential of these immature ILC subsets in the intestine. Based on the gating strategy used to cell sort eNK cells (Fig S5), it is possible that this population was contaminated by CD49a+ stage 4b NK cells. These cells express all the eNK cell markers and therefore could not be discriminated by the sorting strategy employed by the authors, unless CD49a expression was used to exclude them. Could the authors comment on their gating strategy and the purity of their eNK cell subset for their in vitro assays.

Reply: We thank the reviewer for observing that the cell subset that we defined and sorted as eNK cells are very similar to CD49a⁺ stage 4b NK cells in our flow cytometry cluster analyses. Indeed, eNK cells do not express *ITGA1*, the gene for CD49a, and should therefore be defined as CD49a⁻ in our gating strategy. Unfortunately, we did not include CD49a in our gating strategy due to lack of space in the sorting panel, so it is indeed possible that some CD49a⁺ stage 4b NK cells were included in the eNK cell gate. We confirmed that by using data from our more extensive flow cytometry panel (Fig 5C-D) where we observed variable CD49a expression on eNK cells (Reviewer figure 7) gated as in the functional differentiation experiments (Fig 6).

Reviewer figure 7. Frequency of CD49a⁺ cells among CD127⁺ eNK cells in normal colon and primary CRC tumors.

Importantly, CD49a expression did not differ between the colon and CRC tumor (Reviewer Fig. 7) and we also did not observe any difference in the differentiation capacity of eNK cells (main Fig. 6) that could be attributed to different phenotypes of eNK cells in colon versus CRC tumors. Since stage 4b NK cells are also NK cell precursors we find that the data generated are still informative in terms of determining the capacity of intratumoral eNK cells to generate ILC1 and NK cells. For transparency we have added a sentence regarding this in the results section page 9, lines 258-260. We also excluded CD49a⁺ stage 4b NK cells from the additional experiments that we performed (page 11, lines 313-314), prompted by the reviewer's question about the relevance of our *in vitro* differentiation culture system (see below).

We further agree with the reviewer that *in vivo* experiments would substantiate the observational and *in vitro* findings in our study. One could envisage an experiment where nILCs or eNK cells are injected in mice. However, we would like to point out that the nILC and eNK cells that we identify here have a clear tissue resident profile (Fig. 1G) and while intratumoral nILCs show similarities to circulating ILC precursors and intratumoral eNK cells are similar to circulating CD56^{bright} NK cells (Fig. S1G), we are not sure that circulating cells would give rise to intratumoral nILCs or eNK cells or if they are

derived from local CD34⁺ hematopoietic stem cells, which are located in the intestine (PMID: 25500367 and PMID: 34453879). Also, it is unclear if injection of intratumoral nILCs or eNK cells would lead to their homing and penetration into tumors. Another challenge is the cell numbers since we can typically only sort 10-100 nILCs per tumor sample and finding so few cells back in the mouse would be challenging. For the mouse model to be at all feasible and relevant, human CRC tumor cells would have to be orthotopically implanted in the colon, and then nILCs or eNK cells would have to be delivered directly into, or along with the human tumor. While we are aware of a few labs in the world that have developed mouse orthotopic CRC models, we have yet not identified a lab that has ever injected human (or mouse) tumor cells in the colon wall of NSG mice. This would mean developing an entirely new CRC model. Discussions with a group in Oxford, UK that recently developed an orthotopic CRC model in immunocompetent mice used approximately 6 months to just learn the complex injections needed. In addition, we would still not know if all the differentiation cues in the NSG mouse system are cross-reactive, even if human cytokines are provided. We therefore find this beyond the scope of the current study.

However, since we agree that our *in vitro* system with OP9-DL1 stromal cells does not represent the tumor microenvironment, we developed a novel *in vitro* system with colorectal cancer cells. We cultured nILCs and eNK cells (now sorted as CD49a⁻) with Caco-2 cells (genetically verified by ATCC) and a modified cytokine mix IL-2+IL-7+IL-23 + IL-1 β (as in figure 6) with the addition of IL-15 to enhance NK cell differentiation. These data have been added as Fig. 7, and Fig. S12-13.

The results recapitulate our findings from the OP9-DL1 cultures in that while nILCs generate both IL-22⁺ ILC3-like cells as well as Perforin⁺Granzyme B⁺ ILC1/NK-like cells, eNK cells differentiate preferentially to ILC1/NK-like cells (Fig. 7, and Fig. S12-13). This is an important validation, also because the eNK cells were sorted as CD49a⁻, validating the existence and function of CD49a⁻ eNK cells.

Furthermore, validating our OP9-DL1 findings, we observed no difference in the differentiation capacity between colon and intratumoral eNK cells. However, we also did not detect any difference in intratumoral nILCs to generate ILC1/NK and ILC2-like cells. This contrasted with our findings with OP9-DL1 stromal cells where intratumoral nILCs generated more ILC1/NK and ILC2-like cells (Fig. 6), the former in agreement with their higher expression of NK cell genes such as *KLRC1* (*NKG2A*) as compared to colon nILCs (Fig. 4A). It is possible that the Caco-2 tumor cell microenvironment in combination with IL-15 contributes with signals that compensates for the lower intrinsic ILC1/NK cell capacity of colon nILCs and overrides the ILC2 potential of tumor nILCs. This would indicate that the tumor microenvironment has a key role in shaping the differentiation capacity of nILCs. It is also worth noting that when comparing the OP9-DL1 and Caco-2 co-culture systems, the latter generated fewer cells expressing CD16, IFN- γ and Granzyme K as well as IL-13 from intratumoral nILCs. Again, this could be due to the tumor microenvironment e.g. impact of TGF- β . In support of that, we detected slightly higher CD103 expression, known to be induced by TGF- β , in the Caco-2 co-culture system as compared to the OP9-DL1. While we are eager to experimentally

dissect such mechanisms, that is challenging due to the low numbers of nILCs and eNK cells, and will hence be the subject of future work.

In the Caco-2 co-culture system we were also able to expand our panel of potential receptors that could mediate the cytotoxic activation of ILC1/NK-like cells generated from nILCs and eNK cells. We detected NKp44, NKp46 and NKG2D on ILC1/NK-like cells (Fig. 7 and Fig. S13), giving clues as to how such cells could be activated in the tumor microenvironment.

We have added these data as Fig. 7, and Fig. S12-13 and included text and discussing regarding these findings on page 11, lines 307-332, page 13, line 398 and page 14, lines 416-430.

3. Finally, as mentioned by the authors, ILCs represent yet untapped therapeutic targets and as such, the authors should extend their flow cytometric analyses and map intratumoral ILC infiltration to correlate the proportion of eNK, nILCs and or other identified ILC subsets with disease prognosis, including the likelihood of developing peritoneal metastases. Such analyses would significantly enhance the scope of this manuscript and highlight the clinical importance of these ILC populations in colorectal cancer.

Reply: We thank the reviewer for being positive about the potential therapeutic implications of our findings and we agree that it is important to relate our findings to key clinical outcomes and variables. We agree that our results need to be followed up in a larger patient cohort in the future. We have highlighted this in the section related to strengths and limitations of our study (pages 14-15, lines 437-463).

Our study however represents the most detailed analysis of the ILC-NK cell spectrum of primary CRC to date, and the first ever in CRC-PM tumors. Our scRNAseq dataset contains 23,407 CD127⁺ ILCs, CD56⁺ NK cells as well as Lin⁻CD7⁺CD56⁻CD127⁻ non-conventional ILCs, which, to our knowledge, no previously generated single-cell RNA-sequencing (scRNAseq) data can be compared to. In addition to the 11 patients included for scRNAseq analysis, we included 8 more CRC patients in our flow cytometry validation experiments, and another 24 CRC patients in our functional validation experiments. Hence, data from a total of 43 patients, obtained by three different experimental modalities, univocally showed the existence of immature subsets of ILCs and NK cells. We are therefore confident that the main message of our paper, which is that intratumoral tissue-resident immature ILCs and NK cells can generate differentiated subsets of ILC1/trNK cells in CRC tumors, stands strong.

Still, the reviewer's point is well taken. We therefore re-analyzed our scRNAseq and flow cytometry data and identified a few interesting correlations that we have now added to the manuscript (Fig. S4 and S7). The most consistent finding in both the scRNAseq and flow cytometry data was that nILC and eNK cell frequencies tended to be higher in tumors with less advanced tumor stage (Fig. S4 and S7). These new analyses are mentioned in the text on page 7, lines 186-188 and page 9, lines 249-250.

We additionally observed that patients who relapsed within the first two years following primary colon cancer surgery tended to have higher frequencies of intratumoral ielLC1 CD45RA⁻ and CD49a⁺CD103⁺ St4a NK cells as identified by flow cytometry (Reviewer figure 8, clusters from main Fig 4C-D). However, since only two of the patients relapsed, the statistical power is too low. We have therefore refrained from adding these data to the manuscript.

Reviewer figure 8. Frequency of ILC/NK cell subsets from figure 4 stratified by relapse (n=2) or not (n=5).

As explained in our response to the reviewer’s question 2, we additionally performed an extensive inventory of previous scRNAseq studies of primary CRC to identify one or several datasets to use as an independent validation cohort. Our inventory revealed a study with a large number of patients and decent amounts of cells (see table below for the reviewer’s inspection only; GSE178341) that we reanalyzed and performed cell label transfer analysis (reviewer figure 5 above).

Reviewer Table II. Available scRNAseq data from selected CRC studies.

Accession no	No of primary CRC samples	No of total CD45+ and CD45-cells
GSE221575	9	39,484
GSE178341	62	127,695
GSE146771	18	43,817
GSE178318	6	111,292
GSE225857	4	196,473
GSE232525	Not stated	7,343
GSE242271	5	24,111
GSE200997	16	26,173

We could indeed identify cells matching with several of our clusters (reviewer Fig. 5 above). These cells were found equally prevalent across MMR status, biological sex, tumor location and tumor stages (reviewer Fig. 6 above, note that only 28 of the 62 patients had tumor stage and MMR status data). We have now included these data as supplementary in Fig. S4C-G and they are mentioned in the text on page 7, lines 188-196.

Unfortunately, this dataset (Pelka et al, GSE178341), which is the largest we could find, does not contain information on subsequent development of peritoneal metastases. We have also not found any other scRNAseq dataset with this information. However, using The Cancer Atlas Genome (TCGA) program database, containing whole primary CRC tumor bulk RNA sequencing data, we were able to link the cell clusters we identified here to overall survival. For this we performed deconvolution analysis using InstaPrism which uses the scRNAseq clusters as reference and infers cluster fractions for each sample in the query bulk RNAseq dataset. We then identified tumors with high versus low gene signatures of a particular cluster using the median frequency as the cutoff. Unfortunately, we could not identify any significant differences for any of the cell clusters, including CD16⁺ NK cells, in terms of predicting better or worse overall survival. The examples of nILCs and eNK cells are provided below for the reviewer's inspection only (reviewer Fig. 9).

Reviewer figure 9. Kaplan-Meier curves for overall survival probability versus time (days) for tumors with a high (red) and low (blue) gene signature of A. nILCs or B. eNK cells as defined in main figure 1. The numbers of patients/tumors for each group are stated below the Kaplan-Meier curves.

Reviewer #3 (Remarks to the Author):

This paper provides new insights into the populations of ILCs and NK cells present in CRC-PM by utilising single cell sequencing and flow cytometry. The authors identified putative precursor ILC populations that may be of interest for future research

endeavours. This work is original and provides value to the field. Below are some points to be considered.

1. The statistical tests used in figure 3 and figure 6 are not referenced in the figure legends, nor is a legend for p-value indicators – this information should be included. The methods indicate that only t-tests have been used for statistical comparisons. Particularly when multiple groups are being compared, an omnibus test should be used followed by a posthoc test with adjusted p-values so that significance is not overestimated.

Reply: We thank the reviewer for observing our mistake. We have now added information about the statistical test to figure 3 and 6 legends. We also followed the reviewer's advice to apply more rigorous statistical tests to identify differences between groups. We used Two-way ANOVA followed by a Tukey's multiple comparison test with Tukey correction for figure 3. After correction the only significant differences remaining was for NKp44⁺ ILC3 and HLA-DR⁺ NK cells, being depleted or enriched in the tumors versus colon, respectively (Fig. 3). Still, our differential abundance test, which analyses changes in abundance of cell states rather than clusters, showed enrichment of cells in several clusters of the ILC1/trNK branch (Fig. S3). We have made changes related to this topic on page 7, lines 179-180.

For Figure 6, we performed multiple t-test comparisons and added Holm-Šídák correction, considering adjusted p-values. This reduced the numbers of statistical differences. However, our main findings, which was an increased frequency of IL-13⁺ and perforin⁺ ILCs following culture of intratumoral versus colon nILCs, was still statistically significant. There was also a statistical tendency ($p < 0.1$) for increased expression of CD49a on these cells. We also carefully interpreted what seems to be an increase in granzyme B and reduction of CD16 on intratumoral nILCs after culture but highlighted in the text that these were not statistically different. We have made changes related to this on page 10, lines 277-282. We would like to emphasize that our co-expression analysis (Fig. 6H-I) still supports the increased capacity of intratumoral nILC to generate ILC1/trNK-like cells expressing both cytotoxic proteins (perforin and granzyme B and tissue residency markers CD49a and CD103).

2. An effort should be made to improve the interpretability of figure 1, particularly around the numbering and naming of the clusters. For example, in 1D, the key should be titled 'cluster' instead of 'identity'. A table or otherwise should be included to correlated annotated clusters with original cluster numbers rather than having to infer between UMAP plots. Additionally, cluster 10 appears to have been included as part of Branch 3 despite not clustering as such in 1B.

Reply: We agree with the reviewer and have added cluster numbers to Fig 1E-H to improve interpretability.

Regarding cluster 10, the reviewer is right that it was not grouped with any of the 3 branches during hierarchical clustering. However, it was included in branch 3 because of his similarity with other branch 3 clusters including absence of module 6, 8 and 9 genes expressed in either branch 1 or 2 clusters. We have now explained this on page 5, lines 112-115.

3. The axis labels in 2D should be cleaned up.

Reply: This has been done.

4. It may be useful to include the latent time plots alongside your existing RNA velocity cluster plots to improve interpretability of start and endpoints.

Reply: We have added this to Fig 2F. Changes to the text related to this can be seen on page 6, lines 168-170.

5. Figure S3 should have 'differential' rather than 'diferencial' in the titles.

Reply: We thank the reviewer for observing this misspelling, which has now been corrected.

6. You need to include information about the clustering and annotation method for the flow data in figure 5 e.g. unsupervised clustering followed by confirmation with manual gating for annotation.

Reply: We fully agree and have added this information to the manuscript text on page 9, lines 242-243 and in the figure legend for Fig 5. As suggested by the reviewer we performed manual gating to confirm the annotation which has now been added as Figure S5.

7. You should include the gating strategy used for the flow cytometry in figure 5 as a supplementary figure.

Reply: We have added the manual gating for Figure 5 as Figure S5. Changes to the text related to this can be seen on page 9, lines 242-243.

8. Line 241-244 – be more clear about what comparison you are trying to make here.

Reply: This has been done. Changes to the text related to this can be seen on page 9, lines 269-272.

9. In figure S7B you should include the concentration in your concentration/cell CBA plots. I am not sure it is correct to display your concentrations per cell here when you only know the final cell numbers but haven't measured the evolution of these populations over time. This is also true for your CBA plots in figure 6.

Reply: We agree that data expressed as concentration per cell must be interpreted with caution, and this is also why we additionally provided the raw concentrations per well in Fig S7 (now Fig. S10). To further clarify we have added the concentration units (pg/mL) to Figure 6. As the reviewer can also appreciate, we have measured the majority of the proteins by both Luminex and FACS to enable robust interpretations.

10. Line 244-246 – There is no significant difference here, be careful not to bias your interpretation.

Reply: We agree and this has been done on page 10, line 275-276

11. Line 253 – there is a significant difference indicated for CD117.

Reply: We have highlighted this on page 10, line 274

12. You should include discussion around (or mention of) organoids as an alternative differentiation platform for ILCs

e.g. <https://doi.org/10.1016/j.celrep.2022.111281>. Could also discuss WNT signalling as it may affect ILC differentiation in CRC tumours.

Reply: We agree that tumor organoids could serve as an attractive approach to study ILC/NK cell differentiation and we have added a section in the discussion regarding this possibility (page 14, lines 421-422).

With regards to Wnt signalling, we have now discussed this in relation to the OP9-DL1 cultures and Notch signaling, as well as in the context of colorectal cancer on page 13-14, lines 403-413.

Authors' response to the reviewer's Comments

Reviewer #1 (Remarks to the Author):

I think the authors have appropriately addressed the comments.

Reply: We thank the reviewer for this comment.

Reviewer #2 (Remarks to the Author):

I would like to thank the authors for their thorough response to my comments. They have addressed all my comments satisfactorily, and I am confident that this manuscript will serve as a strong reference in the field of ILCs in CRC.

I have a few additional minor comments regarding the newly added data.

1. Lines 259-260. Purity of the sorted eNK CD127+ cells. Could the authors indicate in the figure legends that CD49+ stage 4B NK cells represent approximately ~30% [range:0-70%] of eNK cells. This would help readers better interpret the data shown in Figure 6 and Figure S10.

Reply: We thank the reviewer for this comment, additional information has been added to the text line 259 to help interpretation of the figures

2. Line, 319, the authors suggest that TGF- β production by Caco-2 cells impairs the expression of IFN- γ and granzyme K. However, in the OP9-DL1 system, the culture of nILC or eNK cells (+/-stage 4B NK cells) in the presence of TGF- β results in the accumulation of IFN- γ and granzyme K in the supernatant (Fig. S10). Could the authors comment on these potential discrepancies? Given the potential uncertainties regarding the involvement of TGF- β in this phenotype, I recommend softening this claim and possibly removing TGF- β from the sentence.

Reply: Thank you for this comment which highlights an important point. Indeed, the culture systems used were different not only in terms of the cell lines but also in which cytokine cocktail was used. It is therefore difficult to trace exactly which factors influenced IFN- γ and granzyme K expression. We agree with the reviewer that there might be other factors than TGF- β explaining these discrepancies. We have therefore softened the claim related to TGF- β and also highlighted the fact that IL-12 and IL-18 were missing in the Caco-2 cell co-culture (line 322-323).

3. Line 320, the figure number is missing; it should read (Fig. 7F-I).

Reply: Thank you for this, we have corrected the figure number line 324

4. In Fig 7C, E, G, I, K, M, could the authors include representative FACS plots for NKp44+ ILC3s.

Reply: Thank you for this suggestion, corresponding FACS plots for NKp44+ ILC3s have been added in Figure 7.

5. Line 386. There is a typo – it should read IL-1b.

Reply: Thank you for this, we have corrected the typo now line 390

Reviewer #3 (Remarks to the Author):

The revised manuscript and response to reviewers by Mjosberg et al have largely covered the concerns and suggestions I had with one minor point remaining.

1. Regarding the updated statistical analysis, you are now utilising two-way ANOVA but this is not actually indicated in the text. You should include this and also justification for using a parametric test on a relatively small dataset.

Reply: Thank you for this comment. We have modified the statistical analysis paragraph to add the mention of two-way ANOVA before Tukey's multiple comparison and added a justification of why we chose to use a parametric test despite the small dataset lines 697-702.